# EFFICIENT NUMERACY IN LANGUAGE MODELS THROUGH SINGLE-TOKEN NUMBER ENCODINGS

## ABSTRACT

To drive progress in science and engineering, large language models (LLMs) must be able to process large amounts of numerical data and solve long calculations efficiently. This is currently only possible through the use of external tools or extensive reasoning chains, either weakening the numerical representations of LLMs or limiting the length of problems they can solve. We show that frontier LLMs require excessive amounts of reasoning tokens to solve even basic calculations, which is exacerbated by their tokenization strategies that split single numbers into multiple tokens. This motivates the need for efficient and effective single-token number encodings. We introduce a set of desiderata for such encodings and show that existing approaches fail to fulfill them. To address these shortcomings, we propose *BitTokens*, a novel encoding strategy that represents any number as a single token using its IEEE 754 binary floating-point representation. Through extensive experiments we show that our BitTokens allow even small language models to learn algorithms that solve basic arithmetic operations nearly perfectly. This newly gained efficiency could expand the length and complexity of problems language models can solve.

## 1 INTRODUCTION

Many researchers share the common vision that large language models (LLMs) will not only alleviate routine work but also drive scientific and technological innovation. In many fields, such as physics and engineering, solving such complex tasks requires the processing of large amounts of numerical data and extensive calculations. Thus, to aid advancements in these fields, LLMs must possess efficient and effective numeracy, defined as the ability to represent and compute numbers. However, LLMs have historically struggled to solve even basic calculations (AI4Science & Quantum, 2023; Hager et al., 2024; Cui et al., 2025). A growing body of work has thus tasked itself with finding the capabilities and limitations of transformers to perform mathematical operations (Lee et al., 2024; Charton, 2024; Shen et al., 2023; Nogueira et al., 2021; McLeish et al., 2024).

Research to improve the numeracy of LLMs has predominantly focused on two strategies: arithmetic tool use and reasoning chains. Tool-augmented LLMs leverage external calculators or code to bypass the need for internal arithmetic computation (Schick et al., 2023; Qu et al., 2025; Gou et al., 2024; Zhang et al., 2023; He-Yueya et al., 2023; Parisi et al., 2022; Le et al., 2022; Wang et al., 2024b; Gu et al., 2024; Gao et al., 2023). While this approach guarantees the correctness of the calculations, it forces the model to dynamically and correctly identify, construct, and wait on all mathematical operations, introducing non-negligible latency and sources of error. Furthermore, outsourcing all calculations prevents the model from generating and calculating with intermediate numeric representations during a forward pass, restricting the efficiency of latent calculations(Skean et al., 2024; Hao et al., 2025; Lindsey et al., 2025).

Reasoning chains, on the other hand, prompt LLMs to generate logically consistent text, step-by-step, enabling them to solve complex problems by breaking them into smaller parts (Wei et al., 2022; Yao et al., 2023; Wang et al., 2023; Lightman et al., 2023; Snell et al., 2024). This has resulted in large gains on many benchmarks, pushing the frontier of what is possible with LLMs. However, reasoning chains can be very inefficient, sometimes requiring tens of thousands of tokens to solve a single calculation, as illustrated in Figure 1 and shown later. This limits the length and complexity of problems that LLMs can solve due to context window and cost constraints (Kwa et al., 2025).

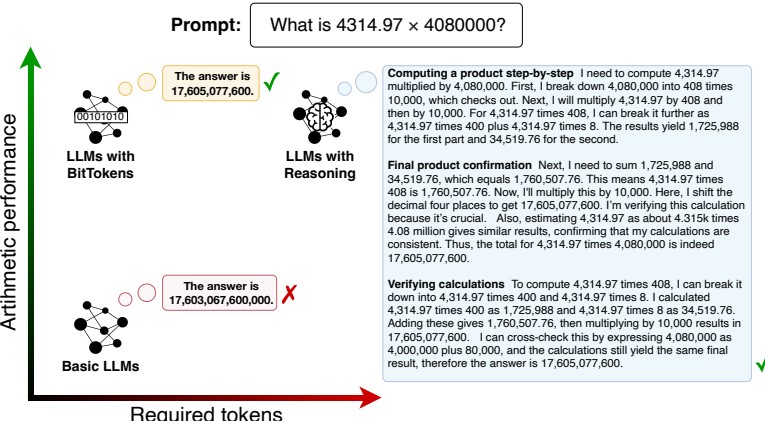

Figure 1: LLMs perform poorly on arithmetic tasks, requiring excessive reasoning tokens to achieve good performance. Our BitTokens tokenization strategy allows language models to solve arithmetic tasks both effectively and efficiently.

We argue that both tooling and reasoning chains are merely crutches, allowing LLMs to solve arithmetic calculations but preventing them from obtaining the intrinsic numeric computation skills required to efficiently solve complex tasks in advanced technical domains. We hypothesize that addressing this problem requires rethinking the way LLMs tokenize and encode numbers. Existing tokenizers treat numbers the same as any other word, meaning they are parsed left to right and split based on a pre-defined vocabulary. This introduces an additional source of token inefficiency, especially in fields such as tabular language models that must ingest tens or hundreds of thousands of numbers (Akhtar et al., 2023; Zhang et al., 2025; Hegselmann et al., 2023; Fang et al., 2024; Liu et al., 2024; Gardner et al., 2024), and is not well suited for comparing and calculating numbers.

A series of works has tried to enhance this encoding step by adding information about the magnitude of numbers via additional tokens (Schwartz et al., 2024) or positional embeddings (McLeish et al., 2024). Others have aimed to address issues of left-to-right token prediction clashing with right-to-left carry-over logic by changing the order or formatting of digits during encoding (Baeumel et al., 2025; Lee et al., 2024; Singh & Strouse, 2024). While all these approaches improve upon standard encoding strategies, they still suffer from the efficiency concerns of single and triple digit tokenization strategies and fail to achieve perfect arithmetic performance.

The most ambitious line of works aims to completely change the way numbers are represented in LLMs, moving from representing individual digits or triplets of digits to encoding each number via a single specialized number token. Han et al. (2022) and Alberts et al. (2024) both employ small neural networks to generate such a number token based on the number's digits and its surrounding textual context. While this improves numeric computational ability, it suffers from similar efficiency drawbacks as conventional tokenization, requires auxiliary networks, and has only been tested for encoding. Ideally one would have a deterministic algorithm to encode numbers into and decode numbers from a fixed structure which would allow LLMs to learn calculation algorithms internally. Accordingly, Golkar et al. (2023) introduce xVal, which scales a learned number token by the numeric value. Zhou et al. (2025) propose FoNE, which encodes a number using sinusoidal functions of different frequencies. While conceptually interesting, we will demonstrate that both methods have fundamental shortcomings that render them ineffective number representations.

With the aim of evaluating and improving both the efficiency and numeracy of LLMs, this work makes the following three major contributions:

- We systematically evaluate the numeric computational ability of eight frontier LLMs across nine different tasks. We find that frontier LLMs strongly rely on reasoning chains with a large number of tokens to solve calculations (see Section 2).

- We develop a set of desiderata for efficient single-token number encodings that allow for networks to learn algorithms that perfectly execute numeric calculations (see Section 3).

In an analysis of recent works, we show that existing single-token strategies fail to fulfill several key desiderata, limiting their potential to be used in LLMs (see Section 4).

- Guided by our desiderata, we propose BitTokens, a novel single-token encoding. Based on the IEEE 754 standard on binary floating-point numbers, BitTokens encode numbers as a sequence of bits representing the sign, exponent, and significand (see Section 5). We demonstrate our method's ability to generate structured, informative representations that allow even small language models to learn algorithms to solve basic arithmetic operations (see Section 6).

## 2 EVALUATION OF NUMERACY IN FRONTIER LLMs

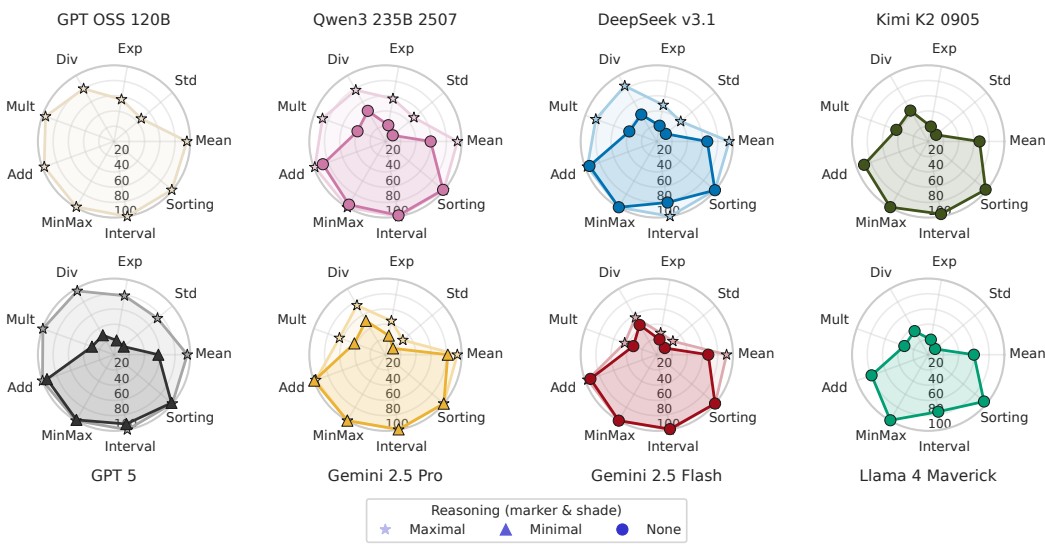

Figure 2: While simple tasks such as addition and comparing numbers are almost perfectly solved by frontier LLMs, other tasks such as multiplication, division, calculating the standard deviation, or exponentiation remain difficult and require extensive reasoning tokens to solve.

**Numeracy benchmarks for language models** Mathematics is a core discipline on which LLMs are evaluated (Chiang et al., 2024; Shao et al., 2024; Zhang et al., 2024; Ahn et al., 2024). However, popular benchmarking datasets like GSM8K, MMLU-Pro, and Numeracy-600K are designed to test mathematical *reasoning* (Cobbe et al., 2021; Mirzadeh et al., 2025; Wang et al., 2024a; Chen et al., 2019). They primarily sample integers from narrow distributions and rely on text-based problems that conflate arithmetic with language understanding. Recently, a few benchmarks have attempted to isolate numeric computations. For example, NumericBench includes an arithmetic operations dataset, although it only tests integers with up to 6 digits and floats with up to 3 decimal places (Li et al., 2025). In contrast, The NumberCookbook tests a wide range of tasks and formats, including number comparisons and arithmetic on integers, floats, fractions, and scientific notation (Yang et al., 2025b). While their benchmark is extensive, we have decided to create our own number dataset for the following reasons: 1. NumberCookbook does not include division between floats, 2. it is missing key operations such as exponentiation, mean, and standard deviation, 3. it includes numbers up to 100 digits, far exceeding the precision capacity of even float64, which only allows for 15-17 digits of precision, and 4. we need to fully control the difficulty distribution of the arithmetic problems and their precision to ensure proper training dynamics.

**Tasks** We design our tasks to isolate individual aspects of numeracy as well as core mathematical operations. The most basic task is to determine whether a numerical comparison of the form $n_1 \{<, >\} n_2$ holds true. To increase the difficulty of number comparisons, we test the ability to: (1) return the minimum or maximum of a list, (2) select the correct interval containing a number from a list of options, and (3) sort a list of numbers in ascending or descending order. To test the calculating

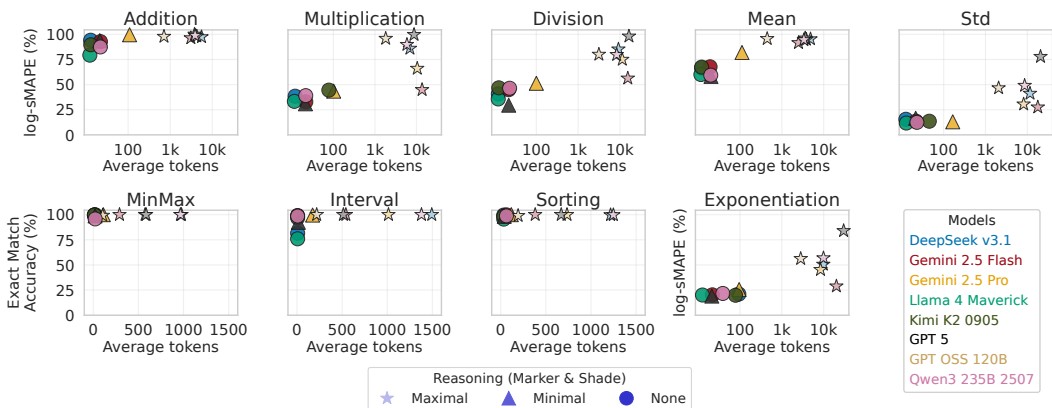

Figure 3: Difficult numeracy tasks such as multiplication, division, exponentiation, and standard deviation can only be solved by frontier models using excessive reasoning tokens.

capabilities of LLMs, we use all basic arithmetic operations including (4) addition / subtraction, (5) multiplication, and (6) division. Finally, we test the multi-step operations (7) exponentiation, (8) mean, and (9) standard deviation. Examples can be found in Appendix A.1.

**Number sampling** We utilize a custom number sampling strategy to ensure that we test a large diversity of inputs. Specifically, all generated numbers $n \in \mathbb{R}$ satisfy $n \in (-10^{15}, -10^{-14}] \cup [10^{-14}, 10^{15})$ and contain at most 15 significant decimal digits, defined as the total number of digits excluding leading and trailing zeros. Number magnitudes, differences between numbers, and precision are all densely and uniformly sampled. Further information can be found in Appendix A.2.

**Evaluation metrics** While exactly matching the true result is the correct answer, it is useful to know the relative error to compare models. To measure the relative error between a prediction $\hat{y}$ and the target $y$, the symmetric mean absolute percentage error (sMAPE) (Flores, 1986) is defined as:

$$\text{sMAPE}(\hat{y}, y) = \frac{|\hat{y} - y|}{|y| + |\hat{y}| + \varepsilon}, \quad \text{with } \epsilon = 10^{-100} \text{ for stability.} \tag{1}$$

While this measures the relative error and is thus well suited for numbers and errors of different magnitudes, the metric decreases exponentially with increasing accuracy in significant figures. An error in the 4th significant digit can thus already achieve a sMAPE on the order of $10^{-4}$ despite the error being potentially very relevant in many real-world contexts. To address this, we propose the log-sMAPE score, which uses the base-10 logarithm of sMAPE divided by the total number of significant digits:

$$\text{log-sMAPE}(\hat{y}, y) = \min\left(1, \frac{-\log_{10}\left(\text{sMAPE}(\hat{y}, y) + \varepsilon\right)}{M}\right) \tag{2}$$

Here $M = 15$ is the maximum number of significant digits tested. In essence, log-sMAPE calculates the fraction of significant figures correctly predicted until the first error. Thus, a log-sMAPE value of 0.2 indicates that the first 3 out of 15 significant digits are correct. We use this metric in addition to exact match accuracy for the addition, multiplication, division, exponentiation, mean, and standard deviation tasks.

**Results** A radar plot of the results is shown in Figure 2 and an overview of the tested models and exact results can be found in Appendix B. It is important to note that we have no way of controlling what the proprietary models (such as GPT 5 and Gemini 2.5) do during the generation of their answer. While we very explicitly instruct them not to use calculators, see that their reasoning chains align with non-calculator use, and they often do not answer correctly, we have no guarantee that they do not use external tools. Their reported results should be seen as an upper bound.

Immediately apparent is the perfect accuracy of almost all models on the basic number comparison tasks of MinMax, Sorting, and Interval, which are effectively solved. When considering the basic

arithmetic tasks, there is a clear separation between high reasoning and non- or minimal-reasoning models. While both perform excellently on addition and mean calculation, all non-reasoning models perform very poorly on multiplication, division, exponentiation, and standard deviation tasks, with none achieving above 60% log-sMAPE. Only once we allow a large number of reasoning tokens do we see improved performance on these tasks. This separation between thinking and non-thinking models is underscored in Figure 3. We see that to improve performance, LLMs use between 5 and 30 thousand tokens to solve a single calculation. Such high token counts for just a single calculation make multi-step calculations and longer reasoning chains infeasible, thus motivating the need for a single-token number representation.

## 3    DESIDERATA FOR SINGLE-TOKEN NUMBER REPRESENTATION STRATEGIES

Our goal is to develop a number encoding strategy that generates representations which both maximize token efficiency and allow for neural networks to learn algorithms that perfectly execute numeric calculations. Such an algorithm or function must be injective and correct over the entire defined numeric input space. It also requires us to consider engineering realities, such as the normalization layers used, the numerical precision of the activation functions, and how gradient-based stochastic optimization learns representations:

**D1 – Token efficiency:** Every number is represented by a single token.

**D2 – Uniqueness:** Each value has exactly one valid encoding, with a unique inverse mapping.

**D3 – Structured:** The encoding geometry reflects numeric order and distance, facilitating the learning of generalizable algorithms.

**D4 – Scale invariance:** The desired range of input magnitudes and precisions can be represented.

**D5 – Normalization:** Encodings are bounded and information preserving under standard normalization functions used in language models (e.g., LayerNorm, RMSNorm).

**D6 – Numerical stability:** Representations remain accurate when using low-precision activations (e.g., FP8).

**D7 – Continuity:** Encodings vary smoothly with the underlying value, making them compatible with gradient-based optimization.

**D8 – Robustness:** Values can be decoded reliably under stochastic noise, allowing for stochastic training.

**D9 – Arithmetic:** Encodings admit learnable algorithms for core mathematical operations.

## 4    ESTABLISHING THE LIMITATIONS OF EXISTING SINGLE-TOKEN STRATEGIES USING OUR DESIDERATA

In this section we will analyze the two existing approaches – xVal (Golkar et al., 2023) and FoNE (Zhou et al., 2025) – for constructing single-token encodings and demonstrate why they are unsuitable using our desiderata. xVal introduces a trainable number token, [NUM], that is scaled by the numerical value being encoded. If a [NUM] token is predicted, the number can be decoded from the final hidden state through a separate number head. This approach satisfies D1 to D3 and D7. However, as shown by the authors, this strategy must map all inputs to the range [-5, 5] to satisfy D5 and avoid layer normalization interfering with the information encoded in the encoding magnitude. This causes it to be extremely limited in the range and precision of numbers it can encode, thus violating D4, D6, and by extension D9.

FoNE encodes a number in a single [NUM] token using Fourier features. Each dimension of the representation corresponds to either a sine or cosine function of base 10 with differing frequencies, each of which encodes the number before being added to a trainable [NUM] token. When a language model predicts a number token, the final hidden state can directly be interpreted as a sinusoidal encoding. The output number is predicted digit by digit through the maximum cosine similarity between each dimension of the final hidden state (corresponding to each digit of the output) with those of the encodings of the numbers in the range $[0, 9]$. This fulfills D1 to D8. In contrast to simply

scaling a [NUM] token, using both sine and cosine functions equally guarantees a constant RMS norm, satisfying D5. Such a digit-wise maximum similarity decoding also allows for robustness to noise, satisfying D8.

The key limitation of sinusoidal encodings is that they are poorly suited for solving certain arithmetic operations with neural networks, most notably multiplication. In order to satisfy condition D9, we require a learnable mapping $OP'$ that computes the encoding, $\xi$, of the result of an operation $OP$ from the encodings of its operands:

$$\forall_{OP} \exists_{OP'}: \quad \xi\big(OP(x_1, \ldots, x_n)\big) = OP'\big(\xi(x_1), \ldots, \xi(x_n)\big) . \tag{3}$$

For addition, such a mapping exists and is computationally trivial.

**Definition 4.1** (**sinusoidal encoding**). *A sinusoidal encoding $\mathcal{F} : \mathbb{R} \mapsto \mathbb{T}^{|\Phi|}$ maps real numbers to a $|\Phi|$-dimensional torus, which forms a compact abelian lie group. Given frequencies $b^\phi$ with base $b > 1$ and $\phi \in \Phi \subset \mathbb{Z}$, then $\mathcal{F}$ is defined as:*

$$\mathcal{F}(x) := \big[\cos(2\pi b^\phi x),\, \sin(2\pi b^\phi x)\big]_{\phi \in \Phi} = \left[e^{i2\pi b^\phi x}\right]_{\phi \in \Phi} \tag{4}$$

**Lemma 4.2** (**additive homomorphism**). *The encoding map $\mathcal{F}$ is a group homomorphism from the additive group of real numbers $(\mathbb{R}, +)$ to the multiplicative torus $(\mathbb{T}^{|\Phi|}, \odot)$, where $\odot$ denotes the element-wise (Hadamard) product.*

*Proof.* The claim follows directly from Euler's formula:

$$\mathcal{F}(x_1 + x_2) = \left[e^{i2\pi b^\phi (x_1 + x_2)}\right]_{\phi \in \Phi} = \left[e^{i2\pi b^\phi x_1} \cdot e^{i2\pi b^\phi x_2}\right]_{\phi \in \Phi} = \mathcal{F}(x_1) \odot \mathcal{F}(x_2) \tag{5}$$

Thus the encoding of the sum of two numbers $x_1$ and $x_2$ can be computed by a simple component-wise multiplication of their respective encodings. $\square$

This homomorphism elegantly transforms addition in the number domain into a simple, local operation in the encoded domain. Notably, this operation does not require carry-over logic. However, for multiplication such a simple mapping does not exist.

**Proposition 4.3** (**non-locality and computational complexity of multiplication**). *Let $\mathcal{X} := \{\varepsilon, \ldots, U\}$ be the set of input numbers with resolution $\varepsilon = b^m$ and choose $\Phi = \{m, \ldots, n\}$ so that $\mathcal{F}$ uniquely encodes the entire number range. Assume each encoding component has a finite precision $P$ (i.e., can represent $P$ distinct states). Suppose there exists an operator $\otimes_\phi : \mathbb{T}^{|\Phi|} \times \mathbb{T}^{|\Phi|} \mapsto \mathbb{T}$ such that $\otimes_\phi(\mathcal{F}(x), \mathcal{F}(y)) := \mathcal{F}_\phi(xy)$ for each output frequency $\phi \in \Phi$. Let $S_\phi^x, S_\phi^y \subseteq \Phi$ be the subsets of input frequencies that $\otimes_\phi$ is required to take as input from $\mathcal{F}(x)$ and $\mathcal{F}(y)$, respectively. Then:*

1. ***Non-locality*** *The operator $\otimes_\phi$ must access at least $d = \mathcal{O}\left(\log_P(U/\varepsilon)\right)$ components from each input vector with $|S_\phi^x|, |S_\phi^y| \geq \lceil \log_P |\mathcal{X}| \rceil$.*

2. ***Computational complexity*** *The operator $\otimes$ must perform a computation functionally equivalent to polynomial multiplication.*

*Proof.* **1. Non-locality.** The proof follows from a counting argument. Assume for contradiction that $|S_\phi^x| < \log_P |\mathcal{X}|$. Then the projection of $\mathcal{F}(x)|_{S_\phi^x}$, which is the only information about $x$ available to $\otimes_\phi$, can represent at most $P^{|S_\phi^x|}$ states. By the pigeonhole principle, there exist $x_1, x_2 \in \mathcal{X}$ with $x_1 \neq x_2$ that are indistinguishable to the operator, i.e., $\mathcal{F}(x_1)|_{S_\phi^x} = \mathcal{F}(x_2)|_{S_\phi^x}$. Let $\Delta := x - x' \neq 0$. The set $Y^* := \{y \in \mathcal{X} : b^\phi \Delta y \in \mathbb{Z}\}$ is a proper subset of $\mathcal{X}$, so we can pick $y^* \in \mathcal{X} \setminus Y^*$. Then $\mathcal{F}_\phi(xy^*) \neq \mathcal{F}_\phi(x'y^*)$, yet $\otimes_\phi$ sees identical inputs from $\mathcal{F}(x)$ and $\mathcal{F}(x')$, which is a contradiction. The operator $\otimes_\phi$ is required by its definition to produce these two different outputs. However, since its inputs for $x_1$ and $x_2$ are identical $(\mathcal{F}(x_1)|_{S_\phi^x} = \mathcal{F}(x_2)|_{S_\phi^x})$, it is forced as a function to produce the same output for both. It cannot satisfy both conditions. Thus, the initial assumption must be false and $|S_\phi^x| \geq \lceil \log_P |\mathcal{X}| \rceil$. The same holds for $|S_\phi^y|$.

**2. Computational complexity.** Multiplication in any positional system is equivalent to the discrete convolution of the coefficients $k$ and $l$, followed by carry propagation.

Write $x = \sum_i k_i b^i$ and $y = \sum_j l_j b^j$ as the sum of their coefficients. Then

$$xy = \sum_\tau (k * l)_\tau b^\tau, \qquad (k * l)_\tau = \sum_{i+j=\tau} k_i l_j. \tag{6}$$

Interpreting the circular sinusoidal component in its interval $[0, 1)$ wrap-around form, we can view the phase vector of the encoding, $\Theta_x \in [0, 1)^{|\Phi|}$, as a linear transformation of the coefficient vector $k$ modulo 1. From the definition of the encoding, the component at frequency $\phi$ accumulates contributions from lower-order coefficients:

$$[\Theta_x]_\phi = \left( \sum_{i < \phi} k_i b^{i-\phi} \right) \bmod 1. \tag{7}$$

This forms a linear system $\Theta_x \equiv Mk \bmod 1$, where $M$ is a lower-triangular mixing matrix with entries $M_{\phi,i} = b^{i-\phi}$ for $i < \phi$ and 0 otherwise. Since the encoding uniquely represents the number range, $M$ is invertible over the relevant domain.

To compute the encoding $\mathcal{F}(xy)$ of the product, the operator must produce the phase vector $\Theta_{xy} \equiv M(k * l) \bmod 1$. We analyze two pathways to achieve this from inputs $\Theta_x$ and $\Theta_y$:

*Pathway A: Disentangle First.* One effectively inverts the mixing matrix to recover coefficients: $k = M^{-1}\Theta_x$ and $l = M^{-1}\Theta_y$. The product is then computed as $\Theta_{xy} = M(M^{-1}\Theta_x * M^{-1}\Theta_y)$. The operation $M^{-1}$ represents the full cost of decoding the sinusoidal representation into positional digits.

*Pathway B: Disentangle Later.* Alternatively, one might attempt to compute the result directly from the entangled phases. Any bilinear operation on the inputs can be expressed via the Kronecker product $\Theta_x \otimes_K \Theta_y$. Substituting the linear forms yields:

$$\Theta_x \otimes_K \Theta_y = (Mk) \otimes_K (Ml) = (M \otimes_K M)(k \otimes_K l). \tag{8}$$

The vector $k \otimes_K l$ contains all cross-terms $k_i l_j$. To construct the convolution $h$, one must sum all subsets of these cross-terms where $i + j = \tau$. However, the term $\Theta_x \otimes_K \Theta_y$ does not provide direct access to $k_i l_j$, only linear combinations weighted by the expanded mixing matrix $M \otimes_K M$.

Isolating the necessary convolution terms from the bilinear expansion requires inverting this mixing process. This removal $(M \otimes_K M)^{-1} = M^{-1} \otimes_K M^{-1}$ of the cross-term redundancies is therefore computationally at least as expensive as the initial decoding $M^{-1}$. Moreover, under finite precision, the condition number satisfies $\kappa(M \otimes_K M) = \kappa(M)^2$ with $\kappa(M) \geq 1$, which means that postponing disentanglement amplifies quantization errors.

Since both pathways require inverting the linear mixing $M$, there exists no shortcut in the sinusoidal domain. Any correct operator $\otimes$ must inherently learn the multi-stage procedure: (1) a non-local decoding of the sinusoidal inputs into internal coefficient sequences for $x$ and $y$, (2) a convolution of these sequences, followed by (3) a carry propagation step, and finally (4) a re-encoding of the result into the sinusoidal format.

$\square$

The preceding proposition demonstrates that performing multiplication in sinusoidal encoding space requires a transformation that is both computationally intensive and prone to precision errors. Any network implementing such an operation is forced to first decode, then calculate, and finally re-encode the encoding. This leads us to conclude that sinusoidal encodings alone are not well-suited as a general purpose number representation.

**Potential improvements to sinusoidal encoding** To address the aforementioned difficulties of performing multiplication in sinusoidal encoding space, we experimented with applying a logarithmic transformation of the input values before encoding them. In logarithmic space, multiplication reduces to component-wise multiplication, following from the identity $\ln(x_1 x_2) = \ln x_1 + \ln x_2$ and Lemma 4.2.

However, this conversion to a logarithmic space trades one homomorphism for another: multiplication becomes Hadamard-linear, but addition no longer admits a simple local rule, which is a well-known drawback of logarithmic number systems (Haselman et al., 2005; Chugh & Parhami, 2013).

Computing $\mathcal{F}_{\log}(x_1 + x_2)$ from $\mathcal{F}_{\log}(x_1)$ and $\mathcal{F}_{\log}(x_2)$ requires learning an effective *exp–sum–log* transformation, reintroducing the same aliasing and inversion difficulties as discussed in Proposition 4.3. Consequently, some operations are easiest to learn in linear space and others in log space. One could simultaneously encode both spaces and use routing mechanisms, such as mixture-of-experts, to enable models to selectively process either encoding and indicate the correct space for decoding. However, our attempts to implement such a solution were unsuccessful.

## 5 BITTOKENS

Guided by the desiderata introduced in Section 3, we propose BitTokens, a novel numeric encoding algorithm. BitTokens uses a dedicated, learnable [NUM] token to which a numeric encoding is added that is based on the IEEE 754 double-precision binary floating-point format (i.e. `float64`) (IEE, 1985; 2008; 2019). The floating-point format writes a signed real value $v$ as

$$v = (-1)^s \left( 1 + \sum_{i=1}^{52} b_{52-i} 2^{-i} \right) \times 2^{E-1023}, \tag{9}$$

with $s \in \{0, 1\}$ denoting the sign bit, $E \in \{0, \ldots, 2047\}$ the 11-bit exponent field offset by 1023, and $b_j \in \{0, 1\}$ for $j = 0, \ldots, 51$ the 52 significand bits.

Each bit is mapped to a dimension of the encoding vector, satisfying D1 and D3. The resulting 64-dimensional number encoding provides a representational range of $[2.23 \times 10^{-308}, 1.8 \times 10^{308}]$ with 15–17 significant decimal digits, satisfying D2 and D4. Additionally, it supports special values such as $\pm 0$, $\pm \infty$, and NaN.

**Encoding construction:** The binary vector of a number can be efficiently constructed via type reinterpretation and bit shifts. To allow for the learning of more efficient division algorithms, we also concatenate the binary representation of the number with a binary representation of its reciprocal value. Scaling the bit vector to $[-1, 1]$ yields unit RMS norm inputs, satisfying D5. We then zero-pad the resulting encoding to match the network's embedding size and then add the vector to the learned [NUM] token embedding. Using raw bit values ensures numerical stability, satisfying D6.

**Decoding and loss computation:** During next token prediction, the language model outputs a [NUM] token to indicate when it is predicting a number. Then the last hidden state that was used to predict the [NUM] token is passed through a dedicated number head, predicting the binary encoding vector. The number head consists of a linear layer followed by a sigmoid activation, with a fixed decision threshold of 0.5 generating the binary vector representation while increasing robustness to prediction noise, thus satisfying D8. During training, computing a regression loss directly on the reconstructed number from the bit vector is infeasible due to the magnitude of potential values as well as the thresholding operations involved in the process. We therefore employ a bit-wise binary cross entropy loss (BCE) and use equal weighting for each dimension. This does not strictly satisfy condition D7, as illustrated by the predictions $\hat{y}_1 = 8 = 0b1000$ and $\hat{y}_2 = 3 = 0b0011$ for the label $y = 7 = 0b0111$. Although $\hat{y}_1$ is numerically closer to $y$, it incurs a higher loss than $\hat{y}_2$. In practice, we find it more important to weigh all bits equally to ensure the more challenging low significance bits receive sufficient attention from the overall loss.

**Arithmetic properties:** Our encoding possesses several properties which ease the learning of mathematical operators, satisfying D9. First, originating from Leibniz's introduction of binary arithmetic (Leibniz (1703)), researchers have developed many efficient algorithms for binary addition and multiplication for modern computing hardware that are compatible with IEEE 754 (Booth, 1951; Wallace, 2006; Brent & Kung, 1982; Kogge & Stone, 2009; Brent & Zimmermann, 2010), Crucially, because each bit is represented independently in our encoding, models can learn algorithms by directly operating upon our BitTokens. As Proposition 4.3 shows, this is not possible with sinusoidal encodings as any algorithm for multiplication must first decode, then compute, and finally re-encode the representations.

Second, the IEEE 754 structure separates values into a logarithmic exponent and a linear significand. The simplest algorithm for addition aligns significands by exponent difference and then adds or subtracts them. Multiplication is also simple as exponents are directly added, significands are multiplied, and the sign is determined by an XOR operation. The direct accessibility of individual bits simplifies the network's ability to learn such algorithms.

Third, bit-wise arithmetic over $\mathbb{Z}_2$ results in coefficient-wise operations reducing to Boolean gates:

$$(x \cdot y) \bmod 2 = x \wedge y, \qquad (x + y) \bmod 2 = x \oplus y. \tag{10}$$

This further simplifies the calculations needed for arithmetic.

Finally, calculations using bits can be parallelized across multiple dimensions of the encoding, reducing the number of sequential steps needed and increasing parameter efficiency for learning algorithms.

## 6 COMPARING BITTOKENS TO OTHER TOKENIZATION STRATEGIES

**Experimental setup** We quantify the impact of different tokenization schemes for numeracy in experiments using nanoGPT-2 models (Jordan et al., 2024a). The use of these small language models allows us to train models from scratch and conduct a diverse set of experiments in a controlled environment, as suggested by Allen-Zhu (2024). A full list of architectural details and hyperparameters is included in Appendix C.1. We use the same number tasks and datasets as in Section 2, generating 30M unique training samples of each task. Simultaneously, we include the FineWeb 10B dataset to model training dynamics with text (Penedo et al., 2024).

We compare BitTokens to traditional single digit and triple digit (subword) tokenizers, as well as xVal and FoNE. To enable xVal to process the large number range in our dataset, we rescale the inputs logarithmically instead of linearly to the range of [-5, 5]. FoNE only encodes the magnitude of the number, thus for negative numbers, a $[-]$ token is added before the absolute number encoding. We use 17 integer and 32 fraction frequencies with base $b = 10$.

**Model training with curriculum learning and dynamic task balancing** Since each task category contains both easy and difficult problems, we observe that the performance of small language models, regardless of their tokenization method, can be improved by strategically choosing the order of training samples. We address this by defining a difficulty metric for each problem and employing a curriculum learning strategy with progressive difficulty thresholds, preview sampling, and adaptive advancement criteria, the details of which can be found in Appendix C.3.

The goal of our experiments is to train a model that can learn all ten tasks simultaneously, a setup we call multi-task. During multi-task training, we find that assigning an equal token budget to each of the nine numeracy plus language tasks yields suboptimal performance. This is due to varying task difficulties and their corresponding imbalanced impact on the overall loss. To address this, we dynamically control the composition of each minibatch. An increased sampling probability is assigned to tasks on which the model performs worse. Additional details and discussion of the dynamic task rebalancing are included in Appendix C.4.

We find that the exponentiation, mean, and standard deviation tasks are intrinsically difficult for small language models. We attribute this to the fact that solving these tasks requires sequential application of multiple arithmetic operations. While dynamic rebalancing improves overall performance, a disproportionate token budget would be assigned to these three tasks, preventing convergence on the remaining seven tasks as shown in appendix D.3. For this reason, we conduct multi-task experiments on the seven comparison and single-step calculation tasks plus text. For exponentiation, mean, and standard deviation, we report the solo-task performance.

**Results** The results of our experiments are shown in Figure 4. Multi-task results show that xVal performs poorly on all tasks, primarily due to its limited precision in representing both input and output. While FoNE is able to learn addition, it struggles with multiplication and division as predicted in Section 4. Among the multi-token strategies, single-digit embeddings consistently outperform subword embeddings. Finally, our BitToken method outperforms all other methods and achieves near-perfect performance on comparison and single-step calculation tasks.

Solo-task results indicate that all models struggle with computing standard deviation and exponentiation. Notably, for the mean task, single digit and subword tokenizer methods outperform the single-token methods. We attribute this advantage to the fact that multi-token methods generate answers over multiple forward passes. These models can refine their answers one digit at a time, making it possible to learn iterative algorithms. In contrast, single-token methods need to perform the same work in a single step. While such a built-in iterative behavior can be beneficial in some

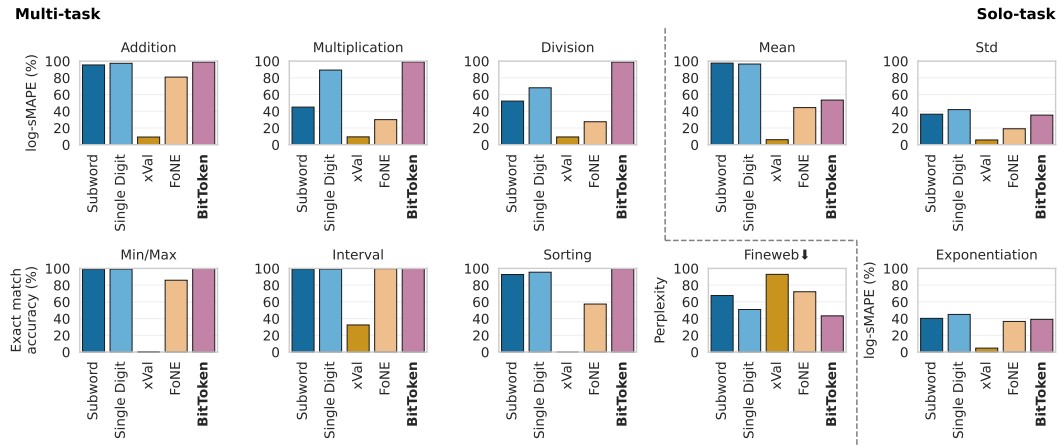

Figure 4: While single digit is the superior multi-token strategy, our BitTokens outperforms it as well as all other single-token strategies on all 7 tasks in the multi-task setting.

cases, it also substantially increases computational costs and generation time for simpler tasks, as shown in table 5. The ablations in Appendix D.3 demonstrate the effectiveness of base-2 encodings over base-10, the benefit of appending the reciprocal, and that curriculum learning substantially boost performance for all methods.

## 7 DISCUSSION AND CONCLUSION

Single-token number representations efficiently encode numerical information in language models. This could reduce the dependency of current frontier LLMs on excessive amounts of reasoning tokens to solve arithmetic problems while simultaneously reducing the amount of tokens needed to encode a single number. We substantially advance this promising line of research by introducing a set of nine desiderata that ensure a number encoding is efficient, trainable, and effective. Based on these, we propose BitTokens, overcoming the drawbacks of existing single-token approaches.

A limitation of our study is that we exclusively evaluate BitTokens on small language models. Although our experiments show that BitTokens integrate well with text during next-token prediction, incorporating them into larger models with broader pretraining data would allow for a more holistic evaluation of their capabilities. Key questions include whether BitTokens enhance the ability of LLMs to memorize and recite numerical facts and the interaction with text in math word problems. Moreover, we do not fully outline the steps required to integrate BitTokens in production-level LLMs. These include efficient strategies to robustly identify and parse numbers in input text, accounting for different arithmetic notations, and ensuring models output the correct level of precision for a given context and respect formatting when appropriate, potentially through the use of dedicated [INT] and [FLOAT] tokens.

While our BitTokens aim to replace excessively long reasoning chains for simple mathematical calculations, reasoning and single-token strategies are not mutually exclusive. Combining them could further improve LLM capabilities on the most complicated problems. By enabling efficient and accurate calculations on intermediate steps, more tokens can be dedicated to logical reasoning, expanding the complexity of problems language models can solve.

## 8 REPRODUCIBILITY STATEMENT

We took several steps to ensure our results are reproducible. Dataset construction is specified in section 2, with the exact number-sampling scheme and the task difficulty metric detailed in appendix A.2 and appendix A.3. The full experimental setup for the BitToken method and all baselines is described in section 6, with model architectures and hyperparameter configurations in appendix C, and additional details for curriculum learning and multi-task training in appendix C.3 and appendix C.4. We include an anonymized, downloadable repository at `https://github.com/AnonymousAuthor553/BitTokens`. This repository includes code for the frontier-model benchmark (including evaluation scripts) and the BitToken method and benchmark. These materials provide the code and configuration files needed to reproduce the reported experiments and regenerate the results referenced in the paper.

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

# A BENCHMARKING DATASET

## A.1 TASKS

### A.1.1 COMPARING NUMBERS

The most elementary numeracy task is comparing two random numbers, $n_1$ and $n_2$. As this task is trivial for modern LLMs, we increase the difficulty to multiple operands.

**Determining the minimum and maximum**

Determining the minimum or maximum of a list of numbers is in essence a series of number comparisons where one number must be held in memory. Each list is comprised of $l$ floats, with $l \in \{2, 3, 4, 5\}$. We use the following phrasings:

> **MinMax phrasing**
>
> ```
> What is the maximum of the list [n₁, n₂]
> What is the minimum of the list [n₁, n₂, n₃, n₄, n₅]
> ```

Our system prompt is:

> **MinMax system prompt**
>
> ```
> You are an expert in numeracy.  For each problem, output only
> valid JSON in this format:
> "answer":  <numeric_answer>
> Do not explain, show steps, or add any extra text.  Do not
> use code blocks to output the answer.
> DO NOT CALL ANY external APIs or use ANY external tool to
> solve the problem.  DO NOT USE a calculator tool.  DO NOT USE
> python.  DO NOT USE Wolfram Alpha.
> The answer must be a single number, exactly as it appears in
> the list.
> ```

We evaluate by calculating the exact string match accuracy of the answer.

**Interval assignment**

Determining if a number falls within an interval can be achieved by a series of pairwise comparisons, where two numbers must be held in memory. Each list is comprised of $l$ floats, with $l \in \{2, 3, 4, 5\}$. We use the following phrasing:

> **Interval phrasing**
>
> ```
> What interval does x=n belong to?  A: x < n₁, B: n₁ <= x <
> n₂, C: n₂ <= x
> ```

Our system prompt is:

> **Interval system prompt**
>
> ```
> You are an expert in numeracy.  For each problem, output only
> valid JSON in this format:
> "answer":  <interval_multiple_choice_answer>
> Do not explain, show steps, or add any extra text.  Do not
> use code blocks to output the answer.
> DO NOT CALL ANY external APIs or use ANY external tool to
> solve the problem.  DO NOT USE a calculator tool.  DO NOT USE
> python.  DO NOT USE Wolfram Alpha.
> The answer must be one of the following:  A, B, C, D, E, F.
> ```

We evaluate by calculating the exact string match accuracy of the answer.

**List sorting**

Sorting a list of numbers is in requires more numbers to be held in memory (depending on the "algorithm" used by the LLM). Each list consists of $l$ floats, with $l \in \{2, 3, 4, 5\}$. We use the following phrasing:

> **Sorting phrasing**
>
> ```
> Sort the list [n₁, n₂, n₃, n₄, n₅] in ascending order.
> Sort the list [n₁, n₂, n₃] in descending order.
> ```

Our system prompt is:

> **Sorting system prompt**
>
> ```
> You are an expert in numeracy.  For each problem, output only
> valid JSON in this format:
> "answer":  <sorted_list>
> Do not explain, show steps, or add any extra text.  Do not
> use code blocks to output the answer.
> DO NOT CALL ANY external APIs or use ANY external tool to
> solve the problem.  DO NOT USE a calculator tool.  DO NOT USE
> python.  DO NOT USE Wolfram Alpha.
> The answer must be a list of numbers.
> ```

We evaluate by calculating the exact string match accuracy of the answer.

### A.1.2 SINGLE-STEP ARITHMETIC

We test both addition and subtraction capabilities in the addition task, but separate multiplication and division as the generation of the multiplicative inverse is not trivial. We use the following phrasing:

> **Single-step arithmetic phrasing**
>
> ```
> What is n₁ + n₂?
> What is n₁ − n₂?
> What is n₁ * n₂?
> What is n₁ / n₂?
> ```

Our system prompt is:

> **Single-step arithmetic system prompt**
>
> ```
> You are an expert in numeracy.  Return exactly one valid JSON
> object in this format:
> "answer":  <numeric_answer>
> Do not explain, show steps, or add any extra text.  Do not
> use code blocks to output the answer.
> DO NOT CALL ANY external APIs or use ANY external tool to
> solve the problem.  DO NOT USE a calculator tool.  DO NOT USE
> python.  DO NOT USE Wolfram Alpha.
> If the answer is not an integer, give it as a decimal (not a
> fraction), rounded to at most 15 significant digits.
> ```

We use $\log$-sMAPE and exact match accuracy as metrics.

### A.1.3 MULTI-STEP ARITHMETIC

The exponentiation task includes both positive and negative exponents as well as float exponents, thus testing both power and root operations. We also test the ability to calculate both the mean and the standard deviation. Calculating the mean requires adding all numbers in the list and then dividing by the length of the list. Calculating the standard deviation requires calculating the mean

of the list, then calculating the average difference between the mean and each element squared, and then taking the square root of that average. We use a single set of number lists for both tasks. Each list is comprised of $l$ floats, with $l \in \{2, 3, 4, 5\}$. We use the following phrasing:

> **Mean and Std Phrasing**
>
> ```
> What is the mean of the list [n₁, n₂, n₃]?
> What is the std of the list [n₁, n₂, n₃, n₄]?
> What is n₁ ^ n₂?
> ```

Our system prompt is:

> **Multi-step arithmetic system prompt**
>
> ```
> You are an expert in numeracy.  Return exactly one valid JSON
> object in this format:
> "answer":  <numeric_answer>
> Do not explain, show steps, or add any extra text.  Do not
> use code blocks to output the answer.
> DO NOT CALL ANY external APIs or use ANY external tool to
> solve the problem.  DO NOT USE a calculator tool.  DO NOT USE
> python.  DO NOT USE Wolfram Alpha.
> If the answer is not an integer, give it as a decimal (not a
> fraction), rounded to at most 15 significant digits.
> ```

We use log-sMAPE and exact match accuracy as metrics.

### A.2 NUMBER SAMPLING

We sample all numbers from the interval $[10^{-14}, 10^{15}) \cup \{0\}$ and ensure that the result is in the same interval. For each task, we sample the operands $n \sim \mathcal{U}(10^x, 10^{x+1})$, where $x \sim \mathcal{U}(-14, 15)$. Similar to Charton (2024), we find that a log-uniform distribution of outcomes in the training set yields the best results. We ensure that all numbers can be represented without loss of information by a float64 tensor and round all answers to at most 15 significant digits. Further, we ensure that the operation of both numbers does not result in vanishing information. For instance, computing the mean of the list $[10^{14}, 10^{-5}, -10^{14}]$ in python would be evaluated as 0 because the intermediate result $10^{14} + 10^{-5}$ cannot be represented given the limited precision of 15-17 digits.

When used by humans, numbers often occur a reduced number of decimal digits, while full precision is used in scientific and engineering environments. To reflect both worlds, we uniformly sample the number of significant digits of the values from a uniform distribution.

For each task, we sample 30M train samples, 10k validation and 10k test samples. In the following, we outline task specific generation parameters.

**MinMax / Interval / Sorting** We draw the list length $l \in \{2, 3, 4, 5\}$ and the magnitude of the mean of all operands from $x \sim \mathcal{U}(-14, 15)$. We then sample operands for each spread $s \in [\min(13, x - 13), \min(x + 2, 13)]$. This increases the number of samples where the numbers share a common prefix and only differ in few digits, thus increasing difficulty. We then draw $m \sim \mathcal{U}(10^x, 10^{x+1})$ and generate a list of numbers $[n_1, \ldots, n_l]$, $n_i \in [m - s, m + s]$ such that $\frac{1}{l} \sum_{i=0}^{l} n_i = m$. We ensure that each list element only has a maximum of $p \sim \mathcal{U}(1, 17)$ significant digits. Because of the limited precision, we cannot guarantee that the mean will be $m$. We therefore draw 1000 lists and take the one that has the closest mean. We ensure that $\sigma \neq 0$, as well as $m \neq 0$ if $l \in \{2, 4\}$ to avoid trivial samples.

For the MinMax task, we randomly choose whether we require the minimum or the maximum of the list. For the sorting task, we randomly choose whether the list should be sorted ascending or descending. For the interval task, we randomly choose a position index $pos \in [0, \ldots, l]$. We then

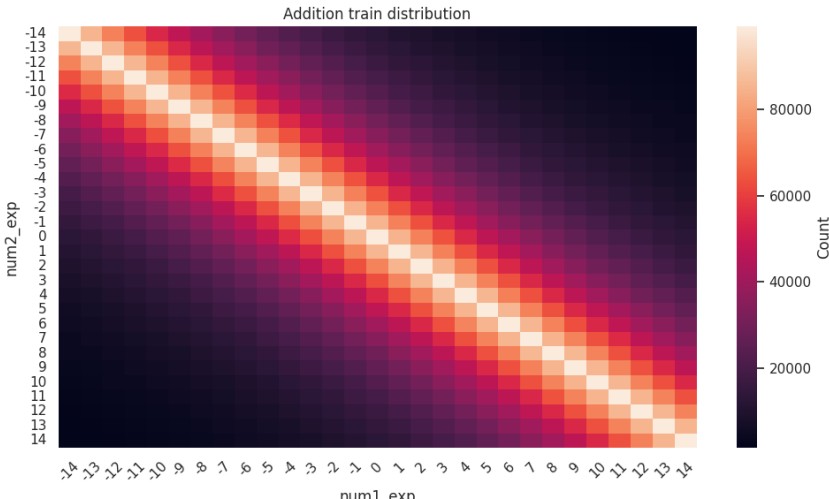

Figure 5: The magnitude distribution of addition pairs in our dataset. Operands with similar exponents are oversampled to increase difficulty.

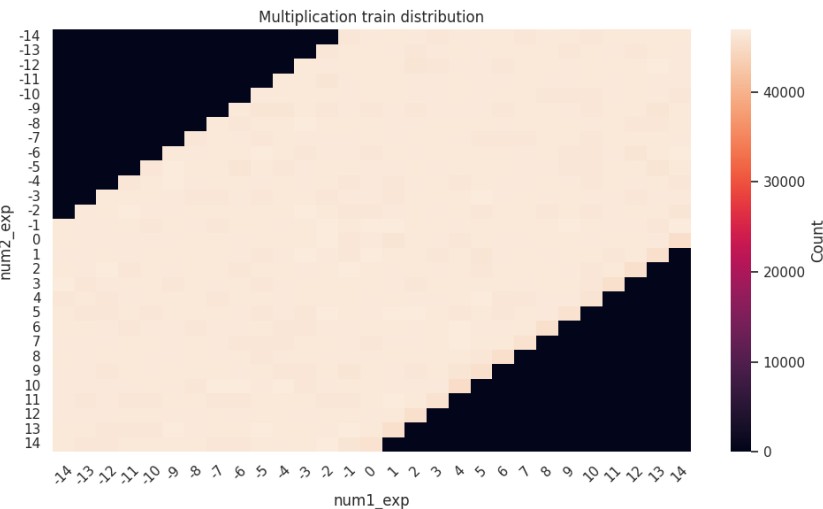

Figure 6: The magnitude distribution of multiplication pairs in our dataset.

draw a reference value from the sorted list $L$:

$$ref \sim \begin{cases} L_0 - 10^s/l & pos = 0 \\ \mathcal{U}(L_{pos-1}, L_{pos}) & 0 < pos < l \\ L_{l-1} + 10^s/l & pos = l \end{cases} \qquad (11)$$

**Addition** The difficulty of addition lies in the carry propagation. If the exponents of the operands are farther apart than the precision limit of float64, the answer to the calculation is simply the larger number. We therefore increase the number of samples where both operands are of similar magnitude (see Figure 5). We sample the combined operand precision $p_{12} \in \mathcal{U}(2, 34)$ from a uniform distribution. We then draw $p_1 \in \text{triangular}(l, l+1, \min(p_{12}-1, 17))$ with $l = \lceil \frac{p_{12}}{2} \rceil$ from a triangular distribution and set $p_2 = p_{12} - p_1$. We then round the operands by $p_1$ and $p_2$ respectively and randomly swap operands. We choose the individual signs using the following schema: (1) both positive in $40\%$ of cases, (2) only one operand negative in $40\%$ of cases and (3) both negative in $20\%$ of cases. We randomly choose an operator $op \in \{+, -\}$.

**Multiplication** We handle precision of the operands and their signs similar to the addition task, however, we do not swap the operands as this could result in the product being out of bounds.

**Division** Similar to addition, we draw the combined precision of the quotient and the divisor and sample their individual precisions. We then compute the dividend from the quotient and divisor. We change signs similar to the addition task, however, we do not swap the operands as this could result in the product being out of bounds.

**Exponentiation** We draw the exponent $x$ from the interval $[-14, 15)$ and increase the number of samples closer to zero. We choose base $b \sim \mathcal{U}(10^x, 10^{x-1})$, choose $\xi \in \left[\left\lceil \frac{10^{-13}}{|x|+1} \right\rceil, \left\lfloor \frac{10^{14}}{|x|+1} \right\rfloor\right]$ and set for all $\xi \notin \{0, 1\}$:

$$exp = \begin{cases} \xi & \xi > 1 \\ 1/\xi & \text{else} \end{cases} \tag{12}$$

This procedure ensures that we only use integer powers and integer roots. If $b > 1$, we randomly set $b$ negative. We also choose the sign of $exp$ randomly.

**Mean and standard deviation** We generate a list similar to the method described for task MinMax but choose $s \in [\min(13, x - 13), \min(x + 17, 13)]$ since here, a smaller spread would simplify the task.

### A.3 PROBLEM DIFFICULTY

Our experiments show that our small scale transformer models benefit from curriculum learning. To maximize sample efficiency, we manually define a difficulty heuristic that is used for sampling. This avoids inefficient loss based selection. Curriculum training is only used for the most difficult tasks multiplication, division, exponentiation, mean and standard deviation.

Most difficulty heuristics are defined using the digit representation of the operands. As a reminder, our BitTokens operate in base-2 instead of the base-10 used by single-digit, triple-digit, xVal, and FoNE. In base $b$, a fraction $\frac{p}{q}$ has a terminating expansion if and only if the prime factors of $q$ (in lowest terms) are all among the prime factors of $b$. Thus, after reducing to lowest terms, for base-2 it must be a power of 2, whereas for base-10 it can a power of 2 or 5 or combination thereof. For example, the fraction $\frac{1}{10}$ terminates in decimal ($\frac{1}{10} = \frac{1}{2^1 * 5^1} = 0.1$) but is infinitely repeating in binary ($0.0\overline{0011}$). This changes the difficulty level of a calculation depending on if the numbers are represented in binary or in decimal.

Our curriculum learning uses the progressive unlocking of more difficult training samples to help learn multiplication and division (see Table 12). Thus, we design two distinct train datasets: one for decimal based number encodings (single digit, triple digit, xVal, FoNE), and one for our binary based encoding (BitTokens) (see Appendix A.2). As humans primarily work in decimal, we sample the difficulty of our test set using the decimal difficulty measure. To account for this distribution shift between train and test data, we switch to the decimal difficulty-sampling distribution for a fixed amount of tokens at the end of our BitTokens training. The total number of tokens used for training is equal for all methods.

In the following, we will describe the difficulty for each task depending on its base. We ensure difficulty $\delta_\tau \in \mathbb{N}_0$.

**Multiplication** Given the fixed-point number representations of the operands in base-2 or base-10, the difficulty $\delta_{\text{Multiplication}}$ is given by the sum of their non-zero digits. This directly correlates with the number of steps involved to solve the task and the numbers' precision.

**Division** Similar to Multiplication, the difficulty $\delta_{\text{Multiplication}}$ is given by the sum of the non-zero digits of dividend, divisor and quotient.

**Mean** For Mean we found the most reliable difficulty metric to be $\delta_{\text{Mean}} = s - x + 15$ with spread $s$ and exponent $x$ of the mean of the list. Intuitively, the difficulty increases for larger spreads. We see a clear drop in performance for all models where $s > exp$, that is, where numbers do no longer share any common prefix.

**Exponentiation / Standard deviation** While the difficulty heuristic for Multiplication, Division and Mean can be derived relatively straightforward, a heuristic for the remaining tasks is a lot harder

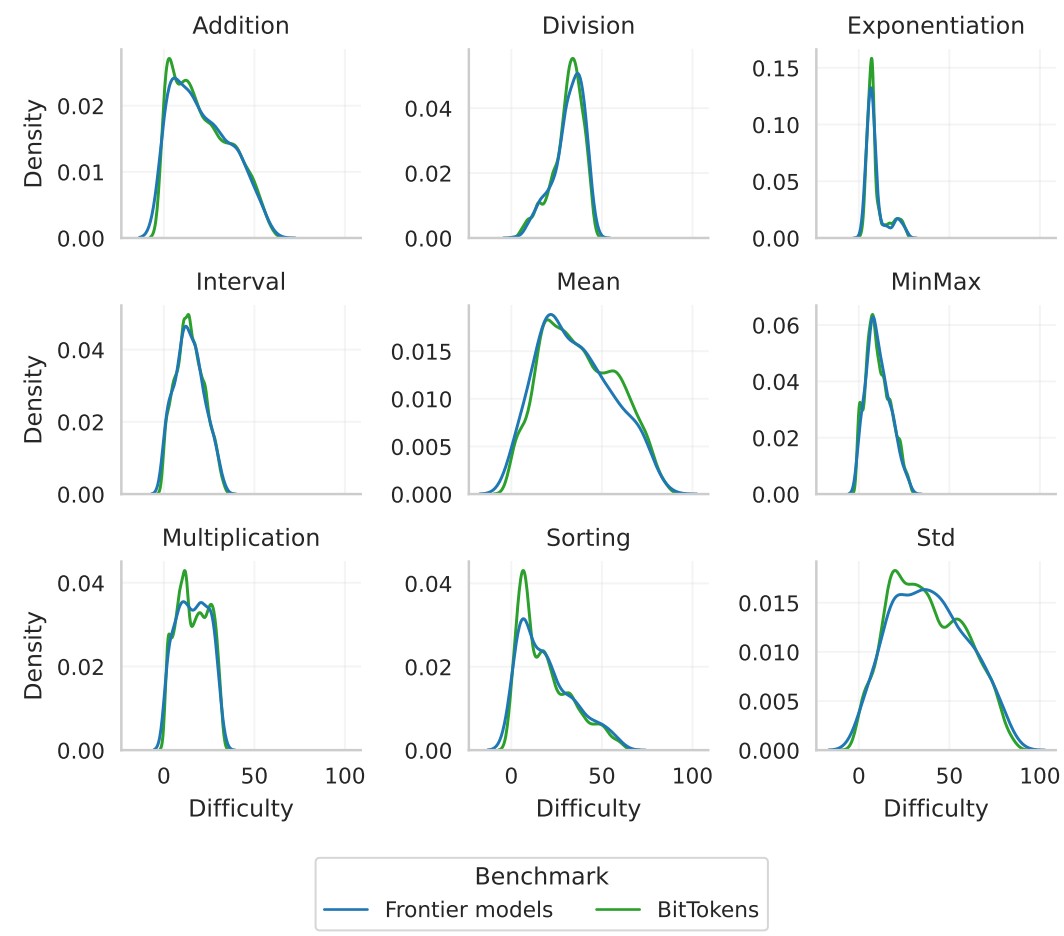

Figure 7: The benchmark for the frontier LLMs is a 500 sample subset of the BitTokens test, but follows the same difficulty distributions.

to define. This is because of their inherent multi-step nature. Each result can be computed using a large variety of different algorithms. For instance, consider the task of computing the standard deviation of a list $L = \{x_1, \ldots, x_N\}$ with mean $\mu$. Possible solutions include $\sqrt{\frac{1}{N} \sum_{i=1}^{N} (x_i - \mu)}$, $\sqrt{\frac{1}{N} \sum_{i=1}^{N} x_i^2 - \mu^2}$, Welford's online algorithm, and many more. We define heuristics $\delta_{\exp}$ as the product of the number of non-zero digits of each base, exponent, and result. For the lack of better alternatives, we set $\delta_{\text{Std}} = \delta_{\text{Mean}}$.

### A.4 SUBSAMPLING

Due to cost constraints we had to subset the full set of 10,000 test samples when evaluating the frontier LLMs. As shown in Figure 7, the benchmark for the frontier LLMs and our BitTokens follow the same difficulty distribution.

# B  FRONTIER BENCHMARK RESULTS

## B.1  FRONTIER LARGE LANGUAGE MODELS:

We benchmark eight frontier LLMs (DeepSeek-AI, 2024; Comanici et al., 2025; Meta, 2025; Team et al., 2025; OpenAI, 2025; Agarwal et al., 2025; Yang et al., 2025a) – both open and closed source – under maximal and minimal reasoning settings. 'Maximal' reasoning means that we select 'high' reasoning for GPT 5 and GPT OSS 120B, set a thinking budget of 32768 tokens for Gemini 2.5 Pro , and set a budget of 24576 tokens for Gemini 2.5 Flash. All models with permanently enabled reasoning on were also labeled 'Maximal'. 'Minimal' reasoning means that we set reasoning to 'minimal' for GPT 5, and we set a thinking budget of 128 tokens for Gemini 2.5 Pro. Models without reasoning or where it can be completely disabled are denoted as 'None' reasoning models. All models use subword or triple digit tokenization for numbers, except Gemini 2.5 Flash and Gemini 2.5 Pro, which use single digit tokenization.

## B.2  LOG-SMAPE PERFORMANCE

| Model | Reasoning | Add | Mult | Div | Mean | Std | MinMax | Interval | Sort | Exp |
|---|---|---|---|---|---|---|---|---|---|---|
| GPT OSS 120B | maximal | 97.6 | 95.8 | 80.0 | 95.6 | 46.6 | 99.2 | 100.0 | 99.0 | 56.0 |
| Qwen3 235B 2507 | maximal | 98.7 | 88.0 | 77.8 | 94.5 | 48.8 | 100.0 | 100.0 | 99.8 | 56.9 |
| Qwen3 235B 2507 | none | 87.5 | 39.0 | 46.4 | 59.4 | 12.5 | 95.8 | 98.6 | 98.8 | 21.7 |
| DeepSeek v3.1 | maximal | 97.0 | 85.3 | 84.2 | 95.0 | 40.9 | 99.8 | 99.8 | 99.6 | 48.6 |
| DeepSeek v3.1 | none | 94.1 | 38.5 | 40.8 | 66.2 | 15.5 | 99.8 | 81.4 | 99.6 | 21.0 |
| Kimi K2 0905 | none | 89.6 | 44.5 | 46.8 | 67.4 | 13.6 | 99.6 | 96.8 | 98.4 | 19.9 |
| Llama 4 Maverick | none | 79.1 | 33.3 | 35.8 | 60.0 | 11.8 | 99.4 | 76.0 | 95.6 | 20.1 |
| GPT 5 | maximal | 100.0 | 99.5 | 96.7 | 96.1 | 74.4 | 100.0 | 100.0 | 99.8 | 78.6 |
| GPT 5 | minimal | 93.7 | 30.9 | 29.5 | 58.2 | 16.5 | 98.8 | 92.4 | 98.4 | 19.0 |
| Gemini 2.5 Pro | maximal | 97.4 | 64.3 | 74.8 | 94.3 | 29.9 | 100.0 | 100.0 | 99.8 | 45.0 |
| Gemini 2.5 Pro | minimal | 99.6 | 43.4 | 51.3 | 81.9 | 13.0 | 100.0 | 99.6 | 100.0 | 25.5 |
| Gemini 2.5 Flash | maximal | 96.4 | 44.9 | 56.3 | 91.6 | 27.5 | 100.0 | 100.0 | 100.0 | 28.9 |
| Gemini 2.5 Flash | none | 92.8 | 32.8 | 45.2 | 67.6 | 13.6 | 100.0 | 99.2 | 99.8 | 20.3 |

Table 1: Average performance of each frontier model on each task. The performance of addition (Add), multiplication (Mult), division (Div), exponentiation (Exp), standard deviation (Std), and mean are all measured in log-sMAPE. The performance of MinMax, Interval, and Sorting are all measured in exact match accuracy.

## B.3 EXACT MATCH ACCURACY

| Model | Reasoning | Add | Mult | Div | Mean | Std | Exp |
|---|---|---|---|---|---|---|---|
| GPT OSS 120B | maximal | 89.4 | 87.2 | 45.4 | 71.2 | 12.6 | 13.4 |
| Qwen3 235B 2507 | maximal | 92.0 | 64.2 | 47.0 | 71.0 | 12.8 | 10.4 |
| Qwen3 235B 2507 | none | 64.4 | 18.2 | 15.6 | 37.2 | 3.0 | 3.4 |
| DeepSeek v3.1 | maximal | 92.4 | 63.6 | 54.0 | 71.2 | 11.0 | 10.6 |
| DeepSeek v3.1 | none | 74.6 | 20.2 | 16.4 | 42.6 | 3.4 | 3.2 |
| Kimi K2 0905 | none | 70.2 | 23.2 | 21.0 | 44.0 | 2.6 | 2.8 |
| Llama 4 Maverick | none | 50.0 | 15.6 | 13.6 | 34.6 | 2.2 | 2.8 |
| GPT 5 | maximal | 98.4 | 97.8 | 91.4 | 73.8 | 34.4 | 46.8 |
| GPT 5 | minimal | 76.6 | 14.2 | 6.6 | 33.2 | 3.8 | 2.4 |
| Gemini 2.5 Pro | maximal | 91.0 | 42.2 | 48.0 | 67.6 | 8.8 | 8.2 |
| Gemini 2.5 Pro | minimal | 82.2 | 20.6 | 21.8 | 55.8 | 4.4 | 3.4 |
| Gemini 2.5 Flash | maximal | 84.0 | 27.8 | 31.6 | 63.4 | 7.0 | 5.6 |
| Gemini 2.5 Flash | none | 72.2 | 13.6 | 17.4 | 40.8 | 2.8 | 2.2 |

Table 2: Average performance of each frontier model on each task. The performance metric for addition (Add), multiplication (Mult), division (Div), exponentiation (Exp), mean, and standard deviation (Std) is exact match accuracy on the first 15 significant digits

## B.4 AVERAGE TOKENS PER PROBLEM

| Model | Reasoning | Add | Mult | Div | Mean | Std | MinMax | Interval | Sorting | Exp |
|---|---|---|---|---|---|---|---|---|---|---|
| GPT OSS 120B | maximal | 708 | 1820 | 3133 | 446 | 2056 | 105 | 218 | 193 | 2815 |
| Qwen3 235B 2507 | maximal | 3899 | 5789 | 8161 | 3359 | 8565 | 965 | 1379 | 1248 | 9889 |
| Qwen3 235B 2507 | none | 22 | 22 | 23 | 20 | 23 | 21 | 7 | 63 | 39 |
| DeepSeek v3.1 | maximal | 5515 | 6921 | 9079 | 4821 | 11486 | 976 | 1490 | 1214 | 9459 |
| DeepSeek v3.1 | none | 13 | 12 | 12 | 12 | 13 | 12 | 7 | 34 | 94 |
| Kimi K2 0905 | none | 13 | 78 | 13 | 12 | 45 | 12 | 7 | 33 | 78 |
| Llama 4 Maverick | none | 12 | 11 | 12 | 12 | 13 | 12 | 7 | 34 | 13 |
| GPT 5 | maximal | 3779 | 8749 | 16165 | 3613 | 19709 | 585 | 514 | 669 | 28086 |
| GPT 5 | minimal | 21 | 21 | 22 | 20 | 21 | 20 | 15 | 39 | 21 |
| Gemini 2.5 Pro | maximal | 4761 | 11901 | 11462 | 4221 | 9281 | 574 | 1013 | 864 | 8748 |
| Gemini 2.5 Pro | minimal | 108 | 101 | 100 | 111 | 165 | 111 | 173 | 117 | 95 |
| Gemini 2.5 Flash | maximal | 3084 | 13698 | 15390 | 2494 | 17751 | 290 | 541 | 382 | 20149 |
| Gemini 2.5 Flash | none | 22 | 22 | 22 | 19 | 22 | 20 | 6 | 63 | 22 |

Table 3: Average number of tokens generated by each frontier model for each problem.

## B.5 PERFORMANCE STABILITY

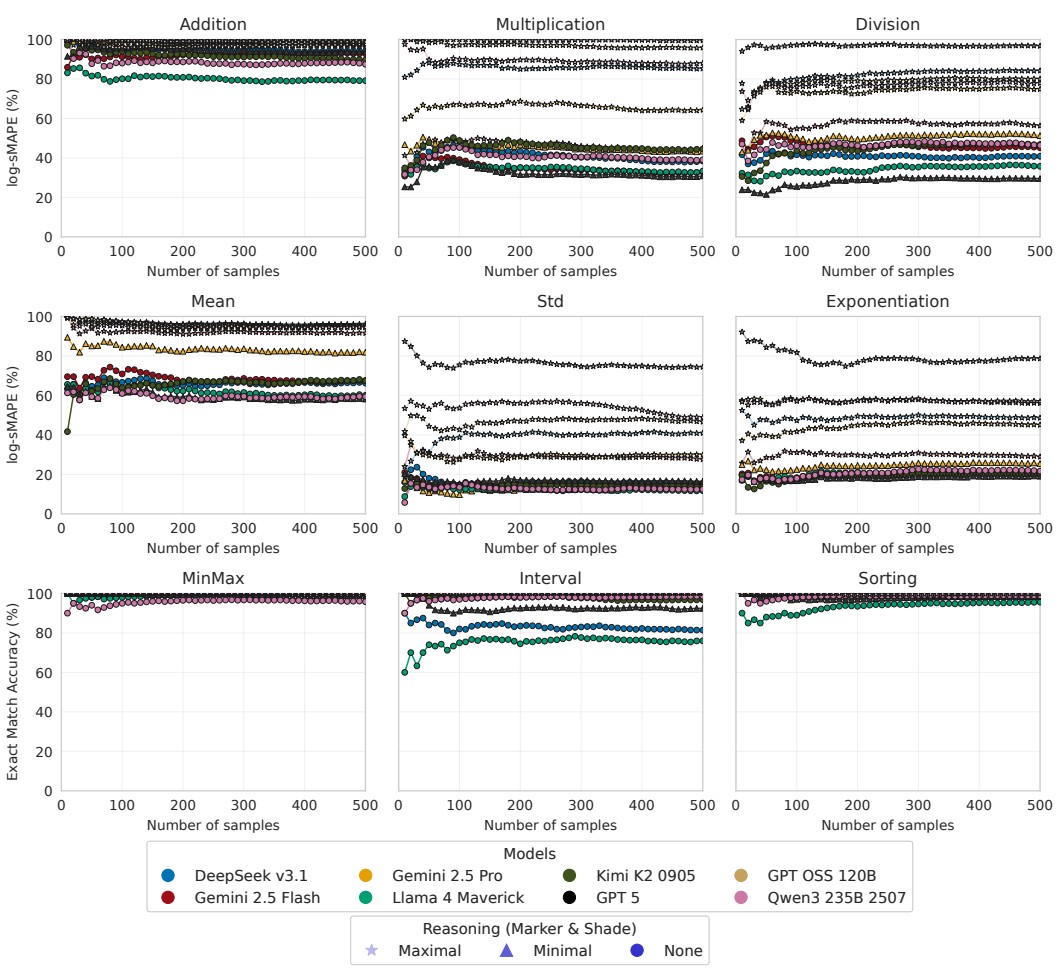

Figure 8: Due to cost constraints we had to subset the full set of 10,000 test samples to 500 samples per task when evaluating the frontier LLMs. We observe performance to already be stable at 500 samples.

# C EXPERIMENTAL SETUP

## C.1 ARCHITECTURE AND HYPERPARAMETER CONFIGURATION

We choose the nanoGPT-2 architecture with 6 layers, 6 attention heads, and an embedding dimension of 768, resulting in a model with 117M parameters. Following Jordan et al. (2024a), we use the Muon optimizer (Jordan et al., 2024b) and modernize the standard architecture of GPT-2 by adding rotary positional embeddings (RoPE) (Su et al., 2024), QK-normalization, and flash attention 2 (Dao, 2024) with sequence packing. The dropout probability is set to 0 to avoid interference with number encodings. Following (Jordan et al., 2024a), all non-scalar parameters in the hidden layers are optimized using the Muon optimizer (Jordan et al., 2024b) with a learning rate of 0.02 and momentum of 0.95. All other parameters are trained with the Adam optimizer (no weight decay) using $\beta_1 = 0.9$ and $\beta_2 = 0.95$. Scalar parameters share the same learning rate of 0.02. The embedding layer weights are untied from the output layer and trained with a learning rate of 0.03. All remaining parameters, including both the language and number heads, use a learning rate of 0.004.

Training is performed with a maximum token budget of 10B tokens, a context length of 1024, and an effective batch size of 192. We employ a cosine learning rate scheduler and allocate 10% of the training tokens for warm-up. Additionally, the Muon optimizer uses 300 momentum warm-up steps. Model performance is evaluated every 32 steps for 2 steps on a small validation set, and the sampling ratio is dynamically adjusted based on current results. After training, we select the checkpoint with the best harmonic mean performance across all tasks.

The training objective is standard next-token prediction, excluding the question tokens from the loss computation. The per-token loss is defined as:

$$\mathcal{L}_{\text{total}} = \text{CrossEntropy}(y_{\text{pred}}, y_{\text{true}}) + 10 \times \mathcal{L}_{\text{num}}(n_{\text{pred}}, n_{\text{true}}),$$

where $\mathcal{L}_{\text{num}}$ is either binary cross-entropy, cross entropy, or mean squared error (MSE), depending on the number encoding strategy.

For testing, we generate tokens by selecting the token with the highest probability.

## C.2 NUMBER PARSING

To parse numerical values from text input, we use the following regexes:

**Python regex for signed values (BitToken / xVal)**
```
r"[-]?(?:(?:0(?!\.[0-9]))|(?:[0-9]*[.][0-9]+)|(?:[1-9][0-9]*))"
```

**Python regex for absolute values (FoNE)**
```
r"(?:(?:0(?!\.[0-9]))|(?:[0-9]*[.][0-9]+)|(?:[1-9][0-9]*))"
```

For values that exceed the numerical range of the `float64` format, the parser splits the numbers into multiple parts, with each part being able to be fully represented by `float64`. This ensures all information is preserved.

The current regex does not parse numbers written in scientific notation as a single value. That is because the use of scientific notation format may be important to the context of the source text. Future work could investigate the use of dedicated tokens for different number formats, such as scientific notation or integer representation.

## C.3 CURRICULUM LEARNING

We observe that small-scale transformer models benefit significantly from curriculum learning when trained on our numerical tasks. To optimize computational efficiency, we implement an automated curriculum scheduler that selects only those samples $\{s_\tau : \delta(s_\tau) \leq \delta_\tau^{\text{frontier}}\}$ whose difficulty does not exceed the current frontier threshold for a given task. For this purpose, we use the difficulty heuristic $\delta(s, \tau)$ defined in Appendix A.3

The initial frontier difficulty is set to $\delta_\tau^{\text{frontier}} = 0.1 \cdot \delta_\tau^{\max}$, and it advances once the model's performance surpasses a predefined threshold, $p(\delta_\tau^{\text{frontier}}, \tau) > \zeta$, where $\zeta = 0.9$. To enhance sample efficiency, we use Equation 16 with $\lambda = 0$ to dynamically compute sampling ratios for each difficulty level based on current performance. This strategy helps mitigate catastrophic forgetting while allocating training tokens to areas of greatest need.

To prevent abrupt shifts in the sample distribution during difficulty transitions, we reserve $20\%$ of the task's token budget for difficulty preview. The corresponding sampling ratios are computed using an exponential decay function:

$$r_{\tau,\delta} = 0.2 \cdot \frac{0.8^{\delta-\delta_\tau^{\text{frontier}}}}{\sum_{\delta>\delta_\tau^{\text{frontier}}} 0.8^{\delta-\delta_\tau^{\text{frontier}}}} \tag{13}$$

To ensure models do not stagnate at lower difficulty levels, we gradually reduce the advancement threshold $\zeta$ over time, starting from the lowest difficulties. This encourages progression to more complex tasks if mastery is not achieved promptly. The dynamic advancement threshold for a given task and difficulty, at any training step and learning rate, is defined as:

$$\zeta(\tau, \delta, \text{step}, \text{lr}) = \min\left(\frac{S}{\text{step}}, \frac{L}{\text{lr}^{\max} - \text{lr}}\right) \cdot \zeta_0 \cdot \frac{\delta_\tau}{\delta_\tau^{\max}} \tag{14}$$

Here, $S$ and $L$ denote the training step and learning rate, respectively, at which the threshold for the highest difficulty begins to decrease. We set this point at $50\%$ of the total training duration to ensure the model has adequate exposure to all difficulty levels in the dataset.

## C.4 MULTITASK LEARNING

Our final goal is to train a model that can solve every numeracy task simultaneously. This is non-trivial since we must optimize nine tasks with varying degrees of difficulty simultaneously. Training is formulated as a multi-task learning problem, aiming to optimize the model's performance $p(\mathcal{T})$ across all tasks $\tau \in \mathcal{T}$ according to the generalized mean:

$$p(\mathcal{T}) = \left(\frac{1}{|\mathcal{T}|} \sum_{\tau \in \mathcal{T}} (p(\tau) + \epsilon)^\lambda\right)^{\frac{1}{\lambda}} \tag{15}$$

The scaling factor $\lambda$ controls how strongly low performing tasks influence the overall score. We set $\lambda = -1$ for the harmonic mean. Since the individual losses can have different magnitudes, we normalize all losses to 10 after warmup. To account for the varying complexities of the tasks, we define a dynamic sample ratio $r_\tau \in (0, 1)$ with $\sum_\tau r_\tau = 1$ for each mini-batch. Given the performance $p(\tau) \in [0, 1]$ of each task $\tau$ and a given time step, we update the sampling ratio by

$$r_\tau^{\text{new}} = \alpha \cdot r_\tau^{\text{old}} + (1 - \alpha) \cdot \frac{(1 - p(\tau))^{1-\lambda}}{\sum_\tau (1 - p(\tau))^{1-\lambda}}, \tag{16}$$

where $\alpha = 0.5$ is a momentum parameter. Intuitively, $\lambda < 0$ leads to more aggressive oversampling of low performing tasks. Appendix D.3 compares the effect of different values of $\lambda$. This multi-task sampling strategy aims to achieve a balanced performance across all tasks without the need for expensive hyperparameter tuning for each setting. We note that this approach becomes less important for larger training budgets.

# D  BITTOKENS BENCHMARK

## D.1  MULTI-TASK RESULTS

| | Metric | Subword | Single Digit | xVal | FoNE | BitToken (Ours) |
|---|---|---|---|---|---|---|
| Min/Max | ↑ Exact match acc | 0.991 | 0.991 | 0.001 | 0.859 | **0.999** |
| Interval | ↑ Exact match acc | 0.993 | 0.992 | 0.324 | 0.997 | **1.000** |
| Sorting | ↑ Exact match acc | 0.927 | 0.955 | 0.000 | 0.574 | **0.998** |
| Add | ↑ log-sMAPE | 0.954 | 0.974 | 0.092 | 0.809 | **0.989** |
| Mult | ↑ log-sMAPE | 0.450 | 0.893 | 0.094 | 0.300 | **0.991** |
| Div | ↑ log-sMAPE | 0.522 | 0.681 | 0.093 | 0.275 | **0.989** |
| Fineweb | ↓ Perplexity | 74.83 | 64.23 | 102.57 | 79.78 | **48.78** |
| Fineweb text only | ↓ Perplexity | 76.56 | 71.85 | 112.03 | 87.01 | **52.54** |

Table 4: This table shows the exact multi-task performance values reported in fig. 4.

| | Subword | Single Digit | xVal | FoNE | BitToken (Ours) |
|---|---|---|---|---|---|
| Min/Max | 34.1 + 8.5 | 60.5 + 16.0 | 11.1 + 2.0 | 11.1 + 2.0 | 11.1 + 2.0 |
| Interval | 45.1 + 2.0 | 27.1 + 2.0 | 15.9 + 2.0 | 15.9 + 2.0 | 15.9 + 2.0 |
| Sorting | 34.0 + 23.1 | 60.0 + 58.0 | 11.1 + 9.5 | 11.1 + 9.5 | 11.1 + 9.5 |
| Add | 21.6 + 9.6 | 32.4 + 16.6 | 5.9 + 2.4 | 5.9 + 2.4 | 5.9 + 2.0 |
| Mult | 21.2 + 10.3 | 32.0 + 17.7 | 5.9 + 2.4 | 5.9 + 2.4 | 5.9 + 2.0 |
| Div | 22.8 + 9.6 | 37.9 + 15.4 | 5.9 + 2.4 | 6.1 + 2.4 | 6.1 + 2.0 |

Table 5: This table shows the mean number of tokens required (input + output) to solve one sample for each task. The predictions were generated by the models trained in the multi-task setting. A large number of output tokens substantially increases generation time, as each output token requires a separate forward pass.

### D.1.1  EXACT MATCH ACCURACY

| | Metric | Subword | Single Digit | xVal | FoNE | BitToken (Ours) |
|---|---|---|---|---|---|---|
| Add | ↑ Exact match acc | 0.926 | 0.962 | 0.002 | 0.767 | **0.977** |
| Mult | ↑ Exact match acc | 0.290 | 0.784 | 0.030 | 0.164 | **0.978** |
| Div | ↑ Exact match acc | 0.379 | 0.514 | 0.013 | 0.136 | **0.982** |

Table 6: Here we report the multi-task exact match accuracy for the single-step calculation tasks.

## D.2 SOLO-TASK RESULTS

|  | Metric | Subword | Single Digit | xVal | FoNE | BitToken (Ours) |
|---|---|---|---|---|---|---|
| Min/Max | ↑ Exact match acc | 0.997 | **1.000** | 0.002 | 0.909 | 0.999 |
| Interval | ↑ Exact match acc | 0.999 | **1.000** | 0.328 | **1.000** | **1.000** |
| Sorting | ↑ Exact match acc | **0.999** | 0.998 | 0.000 | 0.718 | **0.999** |
| Add | ↑ log-sMAPE | 0.999 | **1.000** | 0.090 | 0.851 | 0.998 |
| Mult | ↑ log-sMAPE | 0.564 | 0.964 | 0.096 | 0.328 | **0.985** |
| Div | ↑ log-sMAPE | 0.564 | 0.773 | 0.094 | 0.236 | **0.995** |
| Exp | ↑ log-sMAPE | 0.403 | **0.450** | 0.047 | 0.366 | 0.391 |
| Mean | ↑ log-sMAPE | **0.976** | 0.965 | 0.060 | 0.444 | 0.534 |
| Std | ↑ log-sMAPE | 0.365 | **0.420** | 0.057 | 0.191 | 0.354 |
| Fineweb | ↓ Perplexity | 32.56 | 30.39 | 29.65 | 28.67 | **28.33** |
| Fineweb text only | ↓ Perplexity | 33.13 | 33.29 | 31.62 | 30.65 | **30.19** |

Table 7: This table shows the performance of each model trained on a single task with the same 10B token budged.

### D.3 ABLATIONS

We tested multiple token combination strategies other than our described "Sum" strategy. Product multiplies the [NUM] token with the binary representation. Concat sets the [NUM] token to zero in the 64 binary representation dimensions before adding. Zero pad removes the [NUM] token and inputs the binary representation directly. Weighted passes the binary representation through a linear layer. Weighted + sum passes the binary representation through a linear layer and then adds it to the [NUM] token. Across token combination strategies (Table 8), performance is stable with the simple sum and zero pad strategies performing best.

Changing the numeric base (Appendix D.3) substantially degrades performance, particularly for multiplication and division, highlighting the importance of base choice. We found that switching to layer norm, or changing the number head as shown in appendix D.3, did not meaningfully change performance. Curriculum training (Table 12) yields clear improvements over direct multitask training, especially on arithmetic tasks, confirming its necessity for stable convergence. Appendix D.3 analyzes the effect of including the difficult tasks Exp, Mean, and Std into the multi-task mix. The higher the ratio of these tasks is in the training mix, the more performance degradation can be seen in the remaining tasks. Finally, appending the reciprocal to the encoding improves performance with a negligible performance overhead (Table 11).

|  | Metric | Sum | Product | Concat | Zero Pad | Weighted | Weighted + Sum |
|---|---|---|---|---|---|---|---|
| Min/Max | ↑ Exact match acc | 0.999 | 0.996 | 0.997 | **1.000** | 0.998 | 0.999 |
| Interval | ↑ Exact match acc | **1.000** | **1.000** | **1.000** | **1.000** | **1.000** | **1.000** |
| Sorting | ↑ Exact match acc | 0.990 | 0.967 | 0.966 | **0.992** | 0.983 | 0.989 |
| Add | ↑ log-sMAPE | 0.989 | 0.980 | 0.977 | **0.990** | 0.985 | 0.984 |
| Mult | ↑ log-sMAPE | 0.991 | 0.828 | 0.871 | **0.992** | 0.988 | 0.986 |
| Div | ↑ log-sMAPE | **0.989** | 0.849 | 0.888 | 0.981 | 0.947 | 0.923 |
| Fineweb | ↓ Perplexity | 48.78 | 56.26 | 55.03 | **48.29** | 49.04 | 50.99 |
| Fineweb text only | ↓ Perplexity | 52.54 | 60.68 | 59.32 | **51.91** | 52.73 | 54.89 |

Table 8: Ablation: Performance in the multi-task setting with different combination strategies of the [NUM] token and the binary number representation.

|  | Metric | Base 10 | Layer norm |
|---|---|---|---|
| Min/Max | ↑ Exact match acc | $0.992^{-0.007}$ | $0.997^{-0.003}$ |
| Interval | ↑ Exact match acc | $0.997^{-0.003}$ | $1.000^{+0.000}$ |
| Sorting | ↑ Exact match acc | $0.930^{-0.062}$ | $0.976^{-0.016}$ |
| Add | ↑ log-sMAPE | $0.937^{-0.053}$ | $0.981^{-0.009}$ |
| Mult | ↑ log-sMAPE | $0.526^{-0.466}$ | $0.988^{-0.004}$ |
| Div | ↑ log-sMAPE | $0.437^{-0.544}$ | $0.987^{+0.006}$ |
| Fineweb | ↓ Perplexity | $76.54^{+28.25}$ | $50.10^{+1.81}$ |
| Fineweb text only | ↓ Perplexity | $82.99^{+31.08}$ | $53.94^{+2.03}$ |

Table 9: Ablation: Performance in the multi-task setting with different base encoding and normalization. Deltas to baseline shown as red (degradation) and green (improvement).

|  | Metric | Subword | Single Digit | xVal | FoNE | BitToken |
|---|---|---|---|---|---|---|
| Min/Max | ↑ Exact match acc | $0.988^{-0.003}$ | $\mathbf{0.989}^{-0.002}$ | $0.001^{+0.000}$ | $0.842^{-0.017}$ | $0.988^{-0.012}$ |
| Interval | ↑ Exact match acc | $\mathbf{0.994}^{+0.001}$ | $0.989^{-0.003}$ | $0.308^{-0.016}$ | $0.994^{-0.003}$ | $0.991^{-0.009}$ |
| Sorting | ↑ Exact match acc | $0.923^{-0.004}$ | $\mathbf{0.928}^{-0.027}$ | $0.000^{+0.000}$ | $0.530^{-0.044}$ | $0.911^{-0.081}$ |
| Add | ↑ log-sMAPE | $0.942^{-0.012}$ | $\mathbf{0.951}^{-0.023}$ | $0.097^{+0.007}$ | $0.805^{-0.004}$ | $0.899^{-0.091}$ |
| Mult | ↑ log-sMAPE | $0.436^{-0.014}$ | $\mathbf{0.640}^{-0.253}$ | $0.098^{+0.004}$ | $0.255^{-0.045}$ | $0.582^{-0.410}$ |
| Div | ↑ log-sMAPE | $0.509^{-0.013}$ | $0.482^{-0.199}$ | $0.096^{-0.003}$ | $0.238^{-0.037}$ | $\mathbf{0.596}^{-0.385}$ |
| Fineweb | ↓ Perplexity | $78.08^{+3.25}$ | $74.46^{+0.37}$ | $134.07^{+31.50}$ | $83.46^{+3.68}$ | $\mathbf{74.32}^{+26.03}$ |
| Fineweb text only | ↓ Perplexity | $\mathbf{79.89}^{+5.06}$ | $83.30^{+6.74}$ | $146.83^{+31.50}$ | $91.09^{+4.08}$ | $80.29^{+34.80}$ |

Table 10: Ablation: Performance in the multi-task setting trained with the AdamW optimizer. Deltas to baseline shown as red (degradation) and green (improvement).

|  | Metric | BitTokens w/o reciprocal |
|---|---|---|
| Div | ↑ log-sMAPE | $0.351^{-0.644}$ |

Table 11: Ablation: Removing the reciprocal to the encoding vector severely reduces performance on the division task.

|  | Metric | Subword | Single Digit | xVal | FoNE | BitToken (Ours) |
|---|---|---|---|---|---|---|
| Min/Max | ↑ Exact match acc | $0.990^{-0.001}$ | $0.990^{-0.001}$ | $0.002^{+0.001}$ | $0.874^{+0.015}$ | $0.991^{-0.008}$ |
| Interval | ↑ Exact match acc | $0.990^{-0.003}$ | $0.991^{-0.001}$ | $0.322^{-0.002}$ | $0.997^{0.000}$ | $1.000^{0.000}$ |
| Sorting | ↑ Exact match acc | $0.927^{0.000}$ | $0.949^{-0.006}$ | $0.000^{0.000}$ | $0.622^{+0.048}$ | $0.933^{-0.065}$ |
| Add | ↑ log-sMAPE | $0.950^{-0.004}$ | $0.967^{-0.007}$ | $0.093^{+0.001}$ | $0.813^{+0.004}$ | $0.948^{-0.041}$ |
| Mult | ↑ log-sMAPE | $0.339^{-0.111}$ | $0.484^{-0.409}$ | $0.094^{0.000}$ | $0.285^{-0.015}$ | $0.385^{-0.606}$ |
| Div | ↑ log-sMAPE | $0.433^{-0.089}$ | $0.560^{-0.121}$ | $0.093^{0.000}$ | $0.300^{+0.025}$ | $0.376^{-0.613}$ |
| Fineweb | ↓ Perplexity | $76.92^{+2.09}$ | $72.89^{+8.66}$ | $103.86^{+1.29}$ | $75.99^{-3.79}$ | $77.26^{+28.48}$ |
| Fineweb text only | ↓ Perplexity | $78.75^{+2.19}$ | $81.97^{+10.12}$ | $113.44^{+1.41}$ | $82.85^{-4.16}$ | $82.54^{+30.00}$ |

Table 12: Ablation: Performance in the multi-task setting without curriculum training. Deltas to baseline shown as red (degradation), green (improvement), and gray (no change).

|  | Metric | None | Linear | MLP |
|---|---|---|---|---|
| Min/Max | ↑ Exact match acc | 0.999 | **1.000** | 0.998 |
| Interval | ↑ Exact match acc | **1.000** | **1.000** | **1.000** |
| Sorting | ↑ Exact match acc | **0.994** | 0.992 | 0.989 |
| Add | ↑ log-sMAPE | 0.985 | **0.990** | 0.989 |
| Mult | ↑ log-sMAPE | 0.993 | 0.992 | **0.994** |
| Div | ↑ log-sMAPE | 0.993 | 0.981 | **0.995** |
| Fineweb | ↓ Perplexity | 47.69 | 48.29 | **47.50** |
| Fineweb text only | ↓ Perplexity | 51.08 | 51.91 | **51.06** |

Table 13: Ablation: BitToken performance in the multi-task setting with different number output heads. The encoding is either taking directly from the final hidden layer, transformed by a single linearlayer, or transformed by a 2-layer MLP with ReLU activation.

|  | Metric | harmonic | geometric | arithmetic |
|---|---|---|---|---|
| Min/Max | ↑ Exact match acc | **1.000** | 1.000 | **1.000** |
| Interval | ↑ Exact match acc | **1.000** | **1.000** | **1.000** |
| Sorting | ↑ Exact match acc | 0.992 | 0.995 | **0.997** |
| Add | ↑ log-sMAPE | **0.991** | 0.983 | 0.710 |
| Mult | ↑ log-sMAPE | **0.995** | 0.990 | 0.979 |
| Div | ↑ log-sMAPE | 0.981 | **0.995** | 0.978 |
| Fineweb | ↓ Perplexity | **48.29** | 53.13 | 77.28 |
| Fineweb text only | ↓ Perplexity | **51.91** | 57.25 | 83.85 |

Table 14: Ablation: BitToken performance in the multi-task setting with different task sampling strategies. We compare aggressive oversampling for low performing tasks using the harmonic mean ($\lambda = -1$), moderate oversampling using the geometric mean ($\lambda = 0$), and performance independent sampling using the arithmetic mean ($\lambda = 1$).

|  | Metric | Single Digit with $r_\tau^{\max} =$ | | | | BitToken with $r_\tau^{\max} =$ | | | |
|---|---|---|---|---|---|---|---|---|---|
|  |  | 0.1 | 0.2 | 0.3 | 1 | 0.1 | 0.2 | 0.3 | 1 |
| Min/Max | ↑ Exact match acc | **0.988** | 0.985 | **0.988** | **0.988** | **0.999** | 0.996 | 0.989 | 0.990 |
| Interval | ↑ Exact match acc | **0.991** | 0.989 | 0.990 | **0.991** | **1.000** | **1.000** | 0.999 | **1.000** |
| Sorting | ↑ Exact match acc | **0.927** | 0.897 | 0.909 | 0.911 | **0.980** | 0.963 | 0.920 | 0.922 |
| Add | ↑ log-sMAPE | **0.965** | 0.954 | 0.947 | 0.957 | **0.965** | 0.944 | 0.905 | 0.922 |
| Mult | ↑ log-sMAPE | **0.797** | 0.654 | 0.591 | 0.625 | **0.985** | 0.971 | 0.895 | 0.932 |
| Div | ↑ log-sMAPE | **0.550** | 0.453 | 0.430 | 0.439 | **0.983** | 0.966 | 0.896 | 0.931 |
| Exp | ↑ log-sMAPE | 0.253 | 0.292 | 0.311 | **0.313** | 0.312 | 0.355 | 0.363 | **0.374** |
| Mean | ↑ log-sMAPE | 0.779 | 0.789 | 0.817 | **0.819** | 0.381 | 0.485 | 0.486 | **0.489** |
| Std | ↑ log-sMAPE | 0.214 | 0.237 | 0.357 | **0.358** | 0.214 | 0.306 | 0.304 | **0.324** |
| Fineweb | ↓ Perplexity | **72.29** | 80.50 | 88.37 | 90.37 | **55.22** | 69.22 | 81.44 | 82.26 |
| Fineweb text only | ↓ Perplexity | **81.12** | 90.80 | 100.05 | 101.9 | **59.63** | 75.12 | 88.69 | 89.55 |

Table 15: Ablation: Single digit and BitToken performance in the multi-task setting including difficult tasks with capped maximum task ratio $r_\tau^{\max}$. Best values per model are marked in bold.

