# OpenReview forum: "Efficient numeracy in language models through single-token number embeddings"
_ICLR.cc/2026/Conference — Submitted to ICLR 2026_

### Official Review · Reviewer_THn1 · 2025-10-23

**Soundness:** 3
**Presentation:** 3
**Contribution:** 3
**Rating:** 4
**Confidence:** 4

**Summary:**

The authors in this paper first propose a benchmark to test the numeracy of LLMs, and nine desiderata of the number tokenizer. They analyze two classic previous work xVal and FoNE and find their drawbacks. To solve the problem, they propose their BitTokens which can satisfy all the nine desiderata and can also represent a very large range of numbers. The experiments show that their tokenizer performs better than other tokenizers.

**Strengths:**

1. **Meaningful improvement**: The method detailed analyze the previous mehtods like xVal and FoNE, and propose meaningful improvements over them.
2. Their experiments show that their method outperforms the previous methods on various datasets.
3. The FineWeb dataset is a practical text dataset that can support its effectiveness on real-world tasks.

**Weaknesses:**

1. **Fair comparison between related work**: the related work NumberCookbook contains four different representation of numbers including integer, float, fraction and scientific notation. The float and scientific notation contain large range of numbers. Lines 140-147 incorrectly state about the related works' contribution.
2. **Contribution of the benchmark**: it is not clear what the benchmark adds to the existing related work. It is also not clear that the relation between the results in section 2 and their BitTokens.
3. **The necessary to further justify the nine desiderata**: although the nine desiderata seem to be reasonable, some of them are not well justified.
   1. For example, why "a single token" is required? (D1) A long-standing issue regarding the one-token representation lies in its practical effectiveness. Despite the common emphasis on the computational overhead associated with single-digit tokenizers in certain scenarios, I have yet to see any evidence from efficiency comparisons in real-world tasks that sufficiently demonstrate the importance of this method (including in this article). Considering that, even in fields like physics and finance, the number of tokens for words, spaces, or other formatting elements (such as tables in markdown) will far exceed that of the numbers themselves, with shorter numbers still predominating, the actual benefits of this method remain questionable.
   2. For D3: we need experiment evidence to show that structured representation leads to better performance, where some indirect evidence is insufficient. I am aware that past work on interpretability in smaller models seems to suggest that the model tends to learn this structural form, but this may be strongly correlated with insufficient model size.
   3. For D6: in most practical scenarios,both the token embedding and the activation value will not use low-precision representation, even the model parameters have heavily quantized. Therefore, it is not clear why low-precision representation is necessary.
4. **Reasoning**: Some text-number reasoning tasks are required to further validate the effectiveness of the proposed method. For example, GSM8K, SVAMP, and other benchmarks that require numerical reasoning in addition to basic understanding. The FineWeb dataset can only validate the fitting ability. Whether the token is suitable for math reasoning is not clear.

**Questions:**

1. What is the contribution on the benchmark?
2. Why do you believe each of the nine desiderata is necessary? Is there any experiment evidence?
3. Is there enough motivation to use a special designed tokenizer to represent number, especially when the experiments show that one-digit tokenizer performs also well on most of the tasks.

---

> ### Author Response · Authors · 2025-11-20
> **Rationale for introducing new number dataset and comparison to NumberCookbook (W1, W2, Q1)**
>
> The reviewer correctly points out that the NumberCookbook contains multiple different number representations instead of just integers, and that its float and scientific notation data covers a large range of number magnitudes. This was imprecisely presented in our original submission as the discussion of the related works NumberCookbook and NumericBench were grouped together. We have now changed the corresponding passage in the paper to better reflect NumberCookbook's contributions:
>
> > Recently, a few benchmarks have attempted to isolate numeric computations. For example, NumericBench includes an arithmetic operations dataset, although it only tests integers with up to 6 digits and floats with up to 3 decimal places (Li et al., 2025). The NumberCookbook tests a wide range of tasks and formats, including number comparisons and arithmetic on integers, floats, fractions, and scientific notation (Yang et al., 2025b).
>
> Although very extensive, NumberCookbook has a few shortcomings as it does not include division between floats and is missing key operations such as  mean, standard deviation, and exponentiation. Moreover, our work focuses on integrating generalized float64 arithmetic and number comparison algorithms with language models. We found that this necessitates specific difficulty sampling strategies and curriculum learning, as described in Appendices A.3 and D.2. NumberCookbook includes numbers up to 100 digits, far exceeding the precision capacity of even float64, which only allows for 15-17 digits of precision. Finally, our experimental setup aims to fully control the difficulty distribution of the arithmetic problems and their precision to ensure proper training dynamics. We have added a new paragraph in Section 2, describing these requirements for our arithmetic benchmark:
>
> > While their benchmark is extensive, we have decided to create our own number dataset for the following reasons: 1. NumberCookbook does not include division between floats, 2. it is missing key operations such as exponentiation, mean, and standard deviation, 3. it includes numbers up to 100 digits, far exceeding the precision capacity of even float64, which only allows for 15-17 digits of precision, and 4. we need to fully control the difficulty distribution of the arithmetic problems and their precision to ensure proper training dynamics.

---

> > ### Comment · Reviewer_THn1 · 2025-11-25
> > **Fair comparison between related work**
> >
> > Now I understand why you need a new benchmark. But it is still a question that whether the benchmark has its own value for other researchers. If the answer is no, it should not be considered as one of your contribution. So I suggest just move your benchmark section as a subsection of your experiments like "test dataset" or something.

---

> > > ### Author Response · Authors · 2025-12-03
> > > **Dataset is not presented as a core contribution of this work**
> > >
> > > We would like to emphasize that our number dataset is neither intended to be presented as a core contribution of our work, nor is it meant to be compared to public numeracy benchmarks such as Number Cookbook. In the current manuscript, our description of the dataset is a single paragraph in our frontier model experiments section, ensuring transparency and reproducibility of our experiments. We do not believe that any section of the paper states that the benchmark is a core contribution.

---

> ### Author Response · Authors · 2025-11-20
> **Justification for our desiderata (W3, Q2) (1/2)**
>
> In response to the reviewer asking for additional justification for some of our desiderata, we have expanded the introduction of Section 3. We have reincluded a description of the principles guiding the design of the desiderata, which was previously removed due to space constraints:
>
> > Our goal is to develop a number encoding strategy that generates representations which both maximize token efficiency and allow for neural networks to learn algorithms that perfectly execute numeric calculations. Such an algorithm or function must be injective and correct over the entire defined numeric input space. It also requires us to consider engineering realities, such as the normalization layers used, the numerical precision of the activation functions, and how gradient-based stochastic optimization learns representations.
>
> Furthermore, the reviewer specifically asks for a detailed rationale for the inclusion of D1, D3, and D6.
>
> Concerning D1 - Token efficiency: The tokens saved by using a single-token representation are enormous when working with larger numeric datasets, such as those processed by tabular LLMs [1,2,3,4]. For example, a simple and commonly used financial dataset of monthly macroeconomic indicators, FRED-MD [5], consists of ~50,000 numbers but ~500,000 digits. Considering such data is typically stored in CSV format, the tokens required for table formatting are manageable (essentially a single comma per entry, more or less equaling the count of numbers). Accordingly, processing this data with a single digit tokenizer approximately results in a more than five times larger sequence length compared to using a single-token number encoding. Use of the former will cause the input sequence to exceed the context window of GPT5.1 (272k) and will quickly fill even Gemini 2.5 Pro’s context window of 1M.
>
> We have added this motivation to our introduction:
>
> > We argue that both tooling and reasoning chains are merely crutches, allowing LLMs to solve arithmetic calculations but preventing them from obtaining the intrinsic numeric computation skills required to efficiently solve complex tasks in advanced technical domains. We hypothesize that addressing this problem requires rethinking the way LLMs tokenize and encode numbers. Existing tokenizers treat numbers the same as any other word, meaning they are parsed left to right and split based on a pre-defined vocabulary. This introduces an additional source of token inefficiency, especially in fields such as tabular language models that must ingest tens or hundreds of thousands of numbers (Akhtar et al., 2023; Zhang et al., 2025; Hegselmann et al., 2023; Fang et al., 2024; Liu et al., 2024; Gardner et al., 2024), and is not well suited for comparing and calculating numbers.
>
> Concerning D3 - Structured: Without a structured number encoding, it is essentially impossible for a learning algorithm to generalize to any numbers not included in the training data. For example, if we were to assign a random hash to each unique number and use this as a number encoding, knowing the encoding of the numbers 5.123 and 5.125 would not provide any information on the encoding of the number 5.124. Conversely, a structured encoding enables the training of learning algorithms, since changes in number magnitude result in predictable changes in the encoding. For this reason, BitTokens, FoNE and xVal all introduce structured encodings. We have extended the description of D3 to make this design rationale more clear:
>
> > Structured - The encoding geometry reflects numeric order and distance, facilitating the learning of generalizable algorithms.
>
> Concerning D6 - Numerical stability: There is a clear trend of even frontier LLMs to very aggressively quantize model activations, with DeepSeek V3 natively using mixed FP8 quantization [6]. Thus, the representation must be compatible with FP8 activations limiting the digit precision of tokens to ~2 digits. Our method easily fulfills this desiderata as it requires only a single bit per dimension to represent a number, being fully compatible with any precision. In contrast, encoding methods that require higher precision to differentiate between numbers, such as xVal, are intrinsically incompatible with these models. We have clarified this in the text, changing D6 to:
>
> > Numerical stability: Representations remain accurate when using low-precision activations (e.g., FP8).

---

> > ### Comment · Reviewer_THn1 · 2025-11-25
> > **About numbers not included in the training data.**
> >
> > The authors say: "Without a structured number encoding, it is essentially impossible for a learning algorithm to generalize to any numbers not included in the training data."
> >
> > First, it is not true if you use digit-based tokenizer. In fact, the current digit based tokenizer just has no prior structural information.
> >
> > Then, I don't think memory but not generalize is a really big problem for our current LLMs. Just add all the numbers into the training set is not a too large set compared to the huge website language corpus.

---

> > ### Comment · Reviewer_THn1 · 2025-11-25
> > **Numerical stability**
> >
> > Just in the technique report of Deepseek V3, they say: "For this reason, after careful investigations, we maintain the original precision (e.g., BF16 or FP32) for the following components: the embedding module, the output head, MoE gating modules, normalization operators, and attention operators. "
> >
> > So, I believe the authors should refine their discussion of the numerical stability. In the implementation of DeepSeek-V3, the embedding is just the original numerical and does not suffer the low quantize.

---

> > > ### Author Response · Authors · 2025-12-03
> > > **Clarification of numerical stability and the structure desideratum**
> > >
> > > We would like to clarify the structure requirement once more.
> > > Digit-based tokenizers do indeed encode a structure, as each digit corresponds to a single learned embedding. Because each digit encoding is a separate vector in sequence dimension, the model can adopt a digit-wise latent look-up table (LUT) for operations. For binary operators, such as addition, a LUT for single-digit numbers requires only $10^2$ entries. When moving to three-digit numbers, the LUT size grows to $1000^2$ entries per operator. Several studies analyzing latent spaces suggest that models attempt to impose structure precisely to avoid the prohibitive cost of such LUTs [1,2,3].
> > >
> > > Next we consider the case of single-token encodings, where one token represents the entire number. For float64, this means encoding more than $10^{19}$ distinct values. To learn an operation such as multiplication without generalization, every possible pair of numbers would need to appear in the training set. This is an infeasible requirement given current token budgets.
> > >
> > > Finally, if trivially enumerating all numbers and including them as training data were a practical solution, major companies with access to large-scale computational resources (e.g., Google, OpenAI, Alibaba) would have already adopted it in all likelihood. The fact that they have not, suggests that this approach is not viable. For these reasons, we argue that generalization is not optional but necessary, and achieving it requires a structured number encoding.
> > >
> > >
> > > [1] Zhu, Fangwei, Damai Dai, and Zhifang Sui. "Language models encode the value of numbers linearly." Proceedings of the 31st International Conference on Computational Linguistics. 2025
> > >
> > > [2] Davies, Alex O., Roussel Nzoyem, and Nirav Ajmeri. "Language Models Do Not Embed Numbers Continuously." arXiv preprint arXiv:2510.08009 (2025).
> > >
> > > [3] Zhou, Tianyi, et al. "Pre-trained large language models use fourier features to compute addition." Advances in Neural Information Processing Systems 37 (2024): 25120-25151.

---

> > > ### Author Response · Authors · 2025-12-03
> > > **FP8 computations in DeepSeek-V3**
> > >
> > > While the reviewer correctly points out that in the Deepseek V3 architecture the embeddings are processed in bfloat16 precision before they enter the network, we would like to kindly bring their attention to the paragraph immediately preceding the quote from the technical report, where the authors state:
> > >
> > > > Firstly, in order to accelerate model training, the majority of core computation kernels, i.e., GEMM operations, are implemented in FP8 precision. These GEMM operations accept FP8 tensors as inputs and produce outputs in bfloat16 or FP32. As depicted in Figure 6, all three GEMMs associated with the Linear operator, namely Fprop (forward pass), Dgrad (activation backward pass), and Wgrad (weight backward pass), are executed in FP8.
> > >
> > > As explicitly stated by the authors and also illustrated in Figure 6 of the report, all inputs are immediately cast to FP8. So the embeddings will also immediately be cast to FP8 when entering the network. We would like to make the point that this actually strongly supports our sixth desideratum that all number encodings have to be compatible with low-precision network activations. Any encoding that loses numeric information when its embedding is typecast to lower precisions, is ineffective.

---

> > ### Comment · Reviewer_THn1 · 2025-11-25
> > **Motivation of one-token embedding**
> >
> > The authors support their motivation by the tabular LLM. If so, I believe the scope of the paper should be restricted to the tabular LLM.

---

> > > ### Author Response · Authors · 2025-12-03
> > > **Tabular LLMs are just one of many applications**
> > >
> > > While we believe that tabular LLMs are an important, active, and high impact area of research where our method could be particularly well suited for, this is definitely not the only application and is not the scope of our paper. As we argue throughout the introduction and is recognized across the field, LLMs are increasingly used for scientific, financial, and engineering purposes. However, in all of these disciplines it is essential that LLMs both efficiently (due to context/ cost constraints) and effectively (they must be correct to be useful) process numbers. We firmly believe that our method significantly advances this line of research, as we are maximally token efficient (requiring only a single forward pass) while also learning some generalized algorithm to calculate (achieving near-perfect generalization performance across all float64 numbers on all four elementary arithmetic operations).

---

> ### Author Response · Authors · 2025-11-20
> **Justification for our desiderata (W3, Q2) (2/2)**
>
> As the reviewer asked whether each of the nine desiderata is necessary, we have briefly outlined the rationale for each desiderata in the list below:
>
> D2 - Uniqueness: Each value has exactly one valid encoding, with a unique inverse mapping. If different numbers would encode to the same representation, it would be impossible to reliably decode any encoded number.
>
> D4 - Scale invariance: The desired range of input magnitudes and precisions can be represented. The encoding method should be defined, unique, and effective over a sufficiently large set of numbers so that it can solve advanced tasks in science, or engineering.
>
> D5 - Normalization: All modern deep neural networks use normalization functions, such as LayerNorm or RMSNorm throughout the network, and these must preserve the information in the numeric representation.
>
> D7 - Continuity: Encodings vary smoothly with the underlying value, making them compatible with gradient-based optimization. If the encoding requires non-smooth jumps in values to approximate numbers, this increases the difficulty of training.
>
> D8 - Robustness: Training deep neural networks with stochastic gradient-based optimization intrinsically contains stochastic noise. If small variations in the predicted number encoding strongly impact its decoded value, network training becomes very inefficient at best.
>
> D9 - Arithmetic: Encodings admit learnable algorithms for core mathematical operations. Only through generalizable algorithms learned in transformer weights can we unlock perfect arithmetic in native LLMs.
>
> [1] Fang, Xi, et al. "Large Language Models (LLMs) on Tabular Data: Prediction, Generation, and Understanding-A Survey." Transactions on Machine Learning Research (2024).
>
> [2] Stefan Hegselmann, Alejandro Buendia, Hunter Lang, Monica Agrawal, Xiaoyi Jiang, and David Sontag. Tabllm: Few-shot classification of tabular data with large language models. In International conference on artificial intelligence and statistics, PMLR, (2023): 5549–5581.
>
> [3] Tianyang Liu, Fei Wang, and Muhao Chen. Rethinking tabular data understanding with large language models. In Proceedings of the 2024 Conference of the North American Chapter of the Association for Computational Linguistics: Human Language Technologies (Volume 1: Long Papers) (2024): 450–482.
>
> [4] Xiaokang Zhang, Sijia Luo, Bohan Zhang, Zeyao Ma, Jing Zhang, Yang Li, Guanlin Li, Zijun Yao, Kangli Xu, Jinchang Zhou, et al. Tablellm: Enabling tabular data manipulation by llms in real office usage scenarios. In Findings of the Association for Computational Linguistics: ACL (2025): 10315–10344.
>
> [5] https://www.stlouisfed.org/research/economists/mccracken/fred-databases
>
> [6] Liu, Aixin, et al. "Deepseek-v3 technical report." arXiv preprint arXiv:2412.19437 (2024).

---

> ### Author Response · Authors · 2025-11-20
> **Evaluation on numeric reasoning benchmarks (W4)**
>
> We acknowledge that we have not evaluated the performance of our BitTokens on numeric reasoning benchmarks, such as GSM8k. We believe this to be an essential next step and mention this in our discussion:
>
> > A limitation of our study is that we exclusively evaluate BitTokens on small language models. Although our experiments show that BitTokens integrate well with text during next-token prediction, incorporating them into larger models with broader pretraining data would allow for a more holistic evaluation of their capabilities. Key questions include whether BitTokens enhance the ability of LLMs to memorize and recite numerical facts and the interaction with text in math word problems.
>
> However, we consider such experiments to be out of scope for this study since the amount of language understanding necessary to properly evaluate such math word problems would require the use of more powerful language models around the size of OLMo 2 1B [1]. We estimated that it would cost 26,752 H100 GPU hours to train such a model with our BitTokens from scratch. Similar to other works, such as NumberCookbook, we focus solely on the core arithmetic capabilities of LLMs without any conflation of text understanding and provide strong evidence that our novel BitTokens are worth scaling up due to their excellent performance on arithmetic computations.
>
> [1] Team OLMo, et al. "2 OLMo 2 Furious." arXiv preprint arXiv:2501.00656 (2024)

---

> ### Author Response · Authors · 2025-11-20
> **Motivation for using BitTokens over single-digit tokenization strategies (Q3)**
>
> We believe there is strong motivation to use our BitTokens compared to a single digit tokenizer due to the substantially improved token efficiency. For an extensive discussion on the rationale for more token efficient number encodings, we refer the reviewer to our justification for our Desideratum D1 in response to the reviewer's second question.

---

### Official Review · Reviewer_M71x · 2025-10-24

**Soundness:** 3
**Presentation:** 4
**Contribution:** 3
**Rating:** 6
**Confidence:** 3

**Summary:**

This paper, Efficient Numeracy in Language Models through Single-Token Number Embeddings, investigates why even frontier LLMs struggle with basic arithmetic despite strong reasoning capabilities. The authors argue that the root cause lies in inefficient number tokenization—existing models split numbers into multiple tokens, forcing them to use long reasoning chains or external tools for simple computations.
To address this, the paper introduces BitTokens, a new single-token number encoding scheme based on the IEEE 754 binary floating-point representation. Each number is encoded as a 64-dimensional binary vector representing its sign, exponent, and significand, concatenated with its reciprocal to improve division learning. This design satisfies a set of nine desiderata for efficient, trainable, and numerically stable encodings.
Comprehensive experiments across nine numeracy tasks show that BitTokens enable even small GPT-style models to perform addition, multiplication, and division with near-perfect accuracy—surpassing prior single-token methods such as xVal (value-scaled embeddings) and FoNE (Fourier Number Embeddings).

**Strengths:**

Originality:
 The paper introduces a novel conceptual and technical framework for enhancing numeracy in LLMs through single-token number embeddings, addressing a long-standing inefficiency in numerical reasoning. The proposed BitTokens represent a creative synthesis of ideas from numerical computing (IEEE 754 floating-point representation) and modern tokenization strategies for LLMs. The formalization of nine desiderata for single-token number encodings is conceptually fresh and provides a principled foundation for evaluating future approaches.


Technical quality:
 The work demonstrates strong theoretical and empirical rigor. The authors analyze prior methods (xVal and FoNE) through formal proofs—e.g., showing the additive homomorphism of sinusoidal encodings and their inability to support efficient multiplication—and motivate BitTokens as a solution grounded in established numerical theory. Experimental methodology is solid: the benchmark includes nine carefully controlled numeracy tasks, diverse number ranges, and rigorous evaluation metrics (e.g., log-sMAPE). Results are robust, replicable, and consistently show clear improvements across models and tasks.


Clarity and presentation:
 The paper is very clearly written and well structured. Each section flows logically—from motivation, desiderata, and theoretical analysis to implementation and results. Visualizations (e.g., Figures 1–4) effectively communicate both the inefficiency of reasoning-based numeracy and the improvements achieved by BitTokens. Mathematical formulations (e.g., Lemma 4.2, Proposition 4.3) are carefully explained and accessible to readers with standard ML background.


Significance and impact:
 The contribution is potentially highly significant for the development of numerically capable LLMs. By providing a deterministic, stable, and efficient encoding for numbers, BitTokens could reduce reasoning-token overhead and unlock more efficient arithmetic computation inside general-purpose LLMs. This directly impacts scientific and engineering applications of LLMs and contributes to the broader goal of building models with intrinsic numerical understanding rather than reliance on external tools.

**Weaknesses:**

The method cannot generalize to unseen or longer-digit numbers across tasks, since each number is represented as a unique token rather than a compositional encoding of digits or bits.


The paper does not explain how the model reproduces exact numeric strings (e.g., 1.000, 000101) where format, not value, is important.


The comparative setup with prior work (FoNE, Neural Number Representations) lacks fairness and direct equivalence in optimization and data sampling settings.

**Questions:**

Generalization to longer or unseen numbers:

 Since each number is represented by its own token, the model cannot share parameters across digits or magnitudes. This means if the training data contains only 3-digit numbers, any 6-digit number in an unseen task will have an untrained token embedding. In practice, you cannot train every number on every task. Could you evaluate how BitTokens handle such out-of-distribution magnitudes — for example, models trained on ≤3-digit numbers but tested on ≥6-digit ones for addition, multiplication, or exponentiation?

---

Exact string prediction:

 How does your pipeline preserve numeric formatting when the target string must match exactly (e.g., 1.000, 000101)? Does the [NUM] decoding step reproduce such surface forms, or does it always canonicalize to a float value (e.g., 1 or 101)?

---

Comparison to related methods:

 Please clarify how BitTokens differ conceptually and empirically from Improving LLM Numerical Reasoning with Neural Number Representations (arXiv:2405.17399), which also explores specialized numeric embeddings for arithmetic reasoning.

---

Experimental fairness:

 In your FoNE reproduction, you employ different optimization and data sampling strategies from the original paper. The FoNE work reports ~97% accuracy on 60-digit addition, suggesting that performance differences may arise from optimizer choice, curriculum design, or sampling distribution rather than representational limits. Could you clarify whether you controlled for these factors? Before large-scale training, it would be informative to compare all methods under identical, simple settings to isolate the effects of different training strategy and training data distribution.

---

> ### Author Response · Authors · 2025-11-20
> **Generalization of BitTokens to longer and unseen numbers (W1, Q1)**
>
> BitTokens can represent any float64 number, spanning magnitudes from $\sim 2.225\times10^{-308}$ to $\sim 1.7977\times10^{308}$ with 15-17 digits of precision. While we did not test arithmetic performance on numbers with more than 15 digits of precision, we believe that the range of float64 is sufficient for almost all real-world arithmetic tasks, including those encountered in engineering and science. Nonetheless, should our model encounter numbers with higher precision than float64, it splits them, as now described in Appendix C.2:
>
> > For values that exceed the numerical range of the float64 format, the parser splits the numbers into multiple parts, with each part being able to be fully represented by float64. This ensures all information is preserved.
>
> We would like to emphasize that our benchmark extensively tests generalization to unseen numbers. Naturally, our test set contains many numbers that are not included in the train set. Our method encodes numbers by combining their binary float64 representation with a learned $[\mathtt{NUM}]$ token. This enables our model to generalize to unseen numbers once it has learned some algorithm that utilizes the binary representation. We believe that our model's near perfect addition, subtraction, multiplication, and division performance empirically demonstrate its generalization capabilities.

---

> ### Author Response · Authors · 2025-11-20
> **Handling variable output formatting (W2, Q2)**
>
> The reviewer is correct to point out that required formatting of numbers may depend on the language context. For example LLMs may need to generate numbers with leading zeros or trailing zeros -- formatting which would normally be removed by Python after decoding, and by extension any number encoding.
>
> Our parsing strategy currently handles leading zeros by tokenizing each zero separately to ensure that no information is lost. A possible solution for generating leading and trailing zeros, which we briefly mentioned in Appendix C.2, is to use dedicated $[\mathtt{INT}]$ and $[\mathtt{FLOAT}]$ tokens to allow the network to decide how the generated output should be interpreted. Generating multiple $[\mathtt{INT}]$ tokens after each other could thus generate any format desired. While we leave the exact design and implementation of such a text parsing and postprocessing solution to future work, we have extended the discussion section to outline these concepts:
>
> > Key questions include whether BitTokens enhance the ability of LLMs to memorize and recite numerical facts and the interaction with text in math word problems. Moreover, we do not fully outline the steps required to integrate BitTokens in production-level LLMs. These include efficient strategies to robustly identify and parse numbers in input text, accounting for different arithmetic notations, and ensuring models output the correct level of precision for a given context and respect formatting when appropriate, potentially through the use of dedicated $[\mathtt{INT}]$ and $[\mathtt{FLOAT}]$ tokens.

---

> ### Author Response · Authors · 2025-11-20
> **Ensuring a fair comparison with FoNE (W3, Q4)**
>
> During our study, we took several precautions to warrant a fair evaluation of all baselines, including FoNE. In particular, we relied on the official implementation of FoNE and ensured that its hyperparameters were tuned appropriately for our experiments. Additionally, we have performed two experiments to verify that our FoNE model was trained correctly.
>
> First, we evaluated the final FoNE model which we trained for our float64 addition task, on the 6-digit decimal test set provided by FoNE. Mirroring FoNE's evaluation setup, we then rounded the predictions to 3 decimal digits. The final exact match accuracy was 99.03%, with the few remaining errors occurring in the last digit, similar to the results reported by the authors of FoNE.
>
> Second, we trained a separate FoNE model using our hyperparameters and optimization setup but using their 6-digit number dataset for training, validation and testing. Again, we were able to recreate their reported results, achieving a 100% exact match accuracy with minimal training. We take this as evidence that our training setup does not unfairly disadvantage FoNE. Instead, we believe that the discrepancy in performance in the two studies is caused by our dataset oversampling "difficult" cases through targeted sampling of operands with similar magnitude, as described in Appendix A.2.
>
> We want to point out that FoNE requires several adaptations to handle 60-digit additions that are reported in their study. As the authors explain in the paper's Appendix E, their model cannot natively handle inputs with more than 15 digits of precision. Furthermore, it suffers from numerical inaccuracies due to the conversion from float64 to float16. To solve 60-digit addition tasks, they had to split the input numbers into groups of 5 digits that are then concatenated into a special representation that starkly differs from the one used throughout their other experiments. While we acknowledge that this is an interesting experiment, we argue that such additions are not common in scientific or engineering. Real-world problems in these domains would most likely rely on scientific notation, which is excellently represented by the float64 standard, which we use throughout our paper.

---

> ### Author Response · Authors · 2025-11-20
> **Comparison to Abacus Embeddings (Q3)**
>
> Compared to the Abacus Embeddings, introduced in the paper referenced by the reviewer [1], our BitTokens differ greatly in both methodological rationale and empirical performance. Abacus Embeddings augment a sequence of tokens representing a number by adding an extra positional embedding to each token. As such, they are an extension to the single digit tokenization strategy. Our BitTokens on the other hand replace the entire number with a single token. This token combines a learned representation with the bit encoding of the number, greatly increasing both token efficiency and arithmetic performance.
>
> Concerning empirical performance, the authors of Abacus Embeddings only examine integer addition, sorting, and multiplication. They report near perfect accuracy on the addition and multiplication tasks and sorting accuracy of ~70%, when using input injection throughout the layers and looped transformers, while we achieve 100% under standard architectures. We have extended the existing discussion of the Abaccus Embeddings in our introduction to emphasize these fundamental differences to our proposed method:
>
> > A series of works has tried to enhance this tokenization step by adding information about the magnitude of numbers via additional tokens (Schwartz et al., 2024) or positional embeddings (Mcleish et al., 2024). Others have aimed to address issues of left-to-right token prediction clashing with right-to-left carry-over logic by changing the order or formatting of digits during encoding (Baeumel et al., 2025; Lee et al., 2024; Singh et al., 2024). While all these approaches improve upon standard encoding strategies, they still suffer from the efficiency concerns of single and triple digit tokenization strategies and fail to achieve perfect arithmetic performance.
>
> [1] McLeish, Sean, et al. "Transformers can do arithmetic with the right embeddings." Advances in Neural Information Processing Systems 37 (2024): 108012-108041.

---

### Official Review · Reviewer_QH8n · 2025-10-31

**Soundness:** 3
**Presentation:** 3
**Contribution:** 2
**Rating:** 4
**Confidence:** 4

**Summary:**

This paper proposes BitTokens, a single token number embedding based on IEEE-754 floating-point structure. Each number is encoded into a [NUM] token augmented with a 64-dimensional vector that corresponds to the sign bit, exponent bits, and significand bits. Decoding uses a small number head with a sigmoid, trained with bit-wise BCE. The authors first motivate the need for efficient numeracy by showing frontier LLMs still require very large reasoning traces for basic arithmetic. They then lay out nine desiderata for single token numeric encodings and argue that prior approaches such as xVal and FoNE violate key desiderata for arithmetic, especially multiplication in sinusoidal space. Experiments on small GPT-2 style models trained from scratch compare BitTokens to subword, single-digit, xVal, and FoNE across seven single-step tasks in a multi-task setting, plus three harder solo tasks. BitTokens achieve near-perfect accuracy on comparison and single-step arithmetic, with notably strong multiplication and division, and competitive language modeling perplexity on FineWeb. The paper also introduces a curriculum for numeric tasks and dynamic multi-task sampling.

**Strengths:**

Recasting numeric representation as IEEE-754 bit planes inside a single token is a clean idea that aligns with hardware arithmetic and Boolean operations. The formal desiderata are a helpful framework for comparing encodings.

The paper diagnoses why sinusoidal number tokens struggle with multiplication, showing that any learned operator must effectively decode, convolve, propagate carries, then re-encode. This is a sound argument that matches the empirical results.

On multi-task training with text, BitTokens outperform FoNE and xVal on multiplication and division, and are competitive or better than digit and subword baselines on many tasks. With the proposed setup, BitTokens achieve the best FineWeb perplexity among the compared tokenizers in the multi-task configuration.

The design stays numerically stable and LayerNorm-friendly by scaling bits to unit RMS and using a simple number head, which is attractive for integration.

**Weaknesses:**

1. **Training distribution fairness**. The curriculum introduces an **extra training set** that uniformly samples bit precision to balance difficulty for BitTokens, while evaluation retains decimal difficulty for all methods. This is a non-trivial distribution tweak that seems tailored to BitTokens and is not obviously mirrored for baselines. After investigating the provided codes, I find this obvious in the configs: BitTokens have more config training sets compared to other baselines. The paper should either remove this asymmetry or construct equivalently fair curricula for each tokenizer.

2. **Scope of multi-task justification**. The paper argues that exponentiation, mean, and standard deviation are hard and thus are removed from the multi-task mix and trained as solo tasks, which complicates comparisons and the claim that BitTokens deliver broadly better numeracy under realistic pretraining mixes. A stronger justification and matched compute budgets across tasks are needed.

3. **Precision choice not explored.** The method hard-codes float64, yet many deployments run in float32, bfloat16, or even fp8. The advantages of 64-bit mantissa bits vs smaller formats are not quantified, and the desideratum about low-precision robustness is asserted rather than carefully tested across precisions.

4. **Ablation coverage is narrow.** Ablations cover token combination strategies, base-10 vs base-2, reciprocal concatenation, and curriculum. Missing are precision width, number head variants, normalization treatments, noise robustness, and data mixing ratios. Also these methods are trained with Muon optimizer and whether it gives BitTokens unfair favor is unknown. I would like to see standard AdamW results.

5. **Generalization and downstreams.** Results are on small models trained from scratch. It is still unclear how BitTokens interact with large-scale pretraining and math word problems. The discussion acknowledges this but it limits the strength of the contribution for ICLR.

6. **Language modeling tradeoff depends on setup.** In multi task training with text, BitTokens win FineWeb perplexity. In solo task training, FoNE has the best FineWeb perplexity and BitTokens are slightly worse, which weakens the claim that BitTokens are a drop in replacement for BPE or FoNE in general LLM pretraining.

**Questions:**

1. **FineWeb perplexity comparison.** In Table 4 BitTokens have the best perplexity among baselines in the multi-task setup. Can the authors clarify whether FoNE ever wins on perplexity under any controlled setting, for example when numeracy data is removed or curriculum is disabled, and with matched token budgets and seeds? A small ablation in Table 11 hints at shifts when curriculum is off. Please add a controlled study. Also in Table 8, it shows FoNE achieves the best perplexity on solo tasks trained on FinewebText, does that imply the distribution needed for BitTokens to work are so different from natural number distributions?

2. **Distributional fairness of curricula.** The bit-precision-balanced auxiliary training set seems specific to BitTokens. What is the effect size of this design choice on final arithmetic and perplexity metrics? Please either remove this asymmetry or provide equivalent difficulty balancing for decimal-centric tokenizers. A paired ablation would help. I think this is *very important* for scientific studies.

3. **Why multi-task excludes multi-step tasks.** Excluding exponentiation and std from the shared mix weakens the general claim. Could you include them with a capped sampling ratio and report end-to-end training dynamics and final tradeoffs?

---

> ### Author Response · Authors · 2025-11-20
> **Ensuring a fair experimental setup (W1, Q2)**
>
> We appreciate the reviewer’s careful reading of our text and code, and agree that it is paramount that experimental design ensures a fair comparison of all evaluated methods. However, we believe that there has been a misunderstanding regarding the employed training strategies for our method and the baselines in our study.
>
> While conducting our study, we found that curriculum learning helps small language models to learn multiplication and division, as evidenced by the results presented in Appendix D.3, Table 12. This curriculum learning is based on progressive unlocking of more difficult training samples, and therefore requires the definition of a difficulty measure for each arithmetic problem. In our study we define the computational difficulty as the number of significant digits of the representations. However, this measure depends on whether the input number is represented in binary or decimal. For example, the number 0.1 terminates cleanly in decimal while repeating infinitely in binary ($0.0\overline{0011}$), making it easy to process in decimal but difficult in binary. Consequently, we design two distinct train datasets using the respective difficulty measures: one for decimal based number encodings (single digit, triple digit, xVal, FoNE), and one for our binary based encoding (BitTokens), as described in Appendix A.3.
>
> As humans primarily work in decimal, we sample the difficulty of our test set using the decimal difficulty measure. Models using BitTokens, which have been trained on binary difficulty distributions, are trained on the decimal difficulty-sampling distribution towards the end of training. However, the total number of tokens used for training of our proposed method and the baselines is identical. If anything, this experimental setup disadvantages our proposed method as it has to overcome a distributional shift during training, and is trained and tested on data with different distributions. We have clarified this in Appendix A.3, where we now write:
>
> > Most difficulty heuristics are defined using the digit representation of the operands. As a reminder, our BitTokens work in base-2 instead of the base-10 used by single-digit, triple-digit, xVal, and FoNE. In base $b$, a fraction $\frac{p}{q}$ has a terminating expansion if and only if the prime factors of $q$ (in lowest terms) are all among the prime factors of $b$. Thus, after reducing to lowest terms, for base-2 it must be a power of 2, whereas for base-10 it can be a power of 2 or 5 or combination thereof. For example, the fraction $\frac{1}{10}$ terminates in decimal ($\frac{1}{10}=\frac{1}{2^1*5^1}=0.1$) but is infinitely repeating in binary ($0.0\overline{0011}$). This changes the difficulty level of a calculation depending on if the numbers are represented in binary or in decimal.
> >
> > Our curriculum learning uses the progressive unlocking of more difficult training samples to help learn multiplication and division (see Table 12). Thus, we design two distinct train datasets: one for decimal based number encodings (single digit, triple digit, xVal, FoNE), and one for our binary based encoding (BitTokens) (see Appendix A.2). As humans primarily work in decimal, we sample the difficulty of our test set using the decimal difficulty measure. To account for this distribution shift between train and test data, we switch to the decimal difficulty-sampling distribution for a fixed amount of tokens at the end of our BitTokens training. The total number of tokens used for training is identical for all methods.

---

> > ### Comment · Reviewer_QH8n · 2025-11-21
> > **re: data distribution disparity**
> >
> > I don't think I'm fully sold by the argument. A disparity in training data is a disparity in training data. To ensure fair comparison, you should have all the method training on the same type of data. You should either 1. Train BitTokens on the same data distribution as BPE/xVal/FoNE or 2. Train BPE/xVal/FoNE on the same data distribution as BitTokens. Although I get the authors justifications of using this particular data distribution for BitTokens, but it nullifies the claim that BitTokens > BPE/xVal/FoNE. It only shows BitTokens + Special Data Distribution > BPE/xVal/FoNE + Another Standard Data Distribution.

---

> > > ### Author Response · Authors · 2025-12-03
> > > **Models benefit from training distributions tailored for their respective base**
> > >
> > > Following the reviewer’s suggestion, we conducted an ablation experiment where all models are first trained on data from a binary difficulty distribution before fine-tuning them on data from a decimal difficulty distribution.
> > >
> > > |                           | Subword | Single digit | xVal   | FoNE  | BitToken  |
> > > |---------------------------|---------|--------------|--------|-------|-----------|
> > > | Min/Max                   | 0.992   | 0.987        | 0.001  | 0.833 | **1.000** |
> > > | Interval                  | 0.993   | 0.994        | 0.320  | 0.991 | **1.000** |
> > > | Sorting                   | 0.933   | 0.9423       | 0.0001 | 0.521 | **0.992** |
> > > | Add                       | 0.938   | 0.960        | 0.097  | 0.804 | **0.990** |
> > > | Mul                       | 0.421   | 0.474        | 0.095  | 0.202 | **0.992** |
> > > | Div                       | 0.476   | 0.474        | 0.093  | 0.183 | **0.981** |
> > > | Fineweb                   | 77.27   | 74.45        | 100.84 | 81.96 | **48.29** |
> > > | Fineweb text only         | 79.03   | 83.77        | 110.09 | 89.44 | **51.91** |
> > > | Fineweb numeric text only | 71.27   | 77.02        | 104.32 | 82.66 | **45.87** |
> > >
> > > The results above show that all models perform better when trained on distributions tailored for their respective base. Encodings that use base 10 representations, such as xVal or FoNE, benefit from a uniform training distribution with regard to the decimal difficulty, while our binary model benefits from data from a binary difficulty distribution. We strongly believe that this experiment resolves all remaining concerns regarding the fairness of our experimental design.

---

> ### Author Response · Authors · 2025-11-20
> **Including mean, standard deviation, and exponentiation in multi-task training (W2, Q3)**
>
> During preparation of our study, we have conducted experiments that include mean, standard deviation and exponentiation during multi-task training. As shown in the table below and in the newly added Table 15 in Appendix D.3, we found that including these multi-step operations significantly reduced performance on all evaluated tasks for both BitTokens and single digit tokenization.
>
> | | Metric | Single Digit | | | | BitToken | | | |
> | :--- | :--- | :---: | :---: | :---: | :---: | :---: | :---: | :---: | :---: |
> | | | $r_\tau^\text{max}=0.1$ | $r_\tau^\text{max}=0.2$ | $r_\tau^\text{max}=0.3$ | $r_\tau^\text{max}=1.0$ | $r_\tau^\text{max}=0.1$ | $r_\tau^\text{max}=0.2$ | $r_\tau^\text{max}=0.3$ | $r_\tau^\text{max}=1.0$ |
> | Min/Max | $\uparrow$ Exact match acc | **0.988** | 0.985 | **0.988** | **0.988** | **0.999** | 0.996 | 0.989 | 0.990 |
> | Interval | $\uparrow$ Exact match acc | **0.991** | 0.989 | 0.990 | **0.991** | **1.000** | **1.000** | 0.999 | **1.000** |
> | Sorting | $\uparrow$ Exact match acc | **0.927** | 0.897 | 0.909 | 0.911 | **0.980** | 0.963 | 0.920 | 0.922 |
> | Add | $\uparrow$ $\mathrm{log\textnormal{-}sMAPE}$ | **0.965** | 0.954 | 0.947 | 0.957 | **0.965** | 0.944 | 0.905 | 0.922 |
> | Mult | $\uparrow$ $\mathrm{log\textnormal{-}sMAPE}$ | **0.797** | 0.654 | 0.591 | 0.625 | **0.985** | 0.971 | 0.895 | 0.932 |
> | Div | $\uparrow$ $\mathrm{log\textnormal{-}sMAPE}$ | **0.550** | 0.453 | 0.430 | 0.439 | **0.983** | 0.966 | 0.896 | 0.931 |
> | Exp | $\uparrow$ $\mathrm{log\textnormal{-}sMAPE}$ | 0.253 | 0.292 | 0.311 | **0.313** | 0.312 | 0.355 | 0.363 | **0.374** |
> | Mean | $\uparrow$ $\mathrm{log\textnormal{-}sMAPE}$ | 0.779 | 0.789 | 0.817 | **0.819** | 0.381 | 0.485 | 0.486 | **0.489** |
> | Std | $\uparrow$ $\mathrm{log\textnormal{-}sMAPE}$ | 0.214 | 0.237 | 0.357 | **0.358** | 0.214 | 0.306 | 0.304 | **0.324** |
> | Fineweb | $\downarrow$ Perplexity | **72.29** | 80.50 | 88.37 | 90.37 | **55.22** | 69.22 | 81.44 | 82.26 |
> | Fineweb text only | $\downarrow$ Perplexity | **81.12** | 90.80 | 100.05 | 101.9 | **59.63** | 75.12 | 88.69 | 89.55 |
>
>
> We attribute these findings to our curriculum learning dynamically sampling tokens from different tasks based on their current performance, as described in Appendix C.4. As the performance on the mean, standard deviation and exponentiation task is consistently poor, our sampler selects a disproportionate amount of problems from the respective tasks. The insufficient number of tokens from the other arithmetic operations lead to their incomplete learning within the set token budget.
>
> As suggested by the reviewer, we also explored capping the sampling ratio ($r_\tau$) for the three multi-step tasks. The results for maximum ratios of 0.1, 0.2, 0.3 and 1 (no cap) for each individual task are also included in these tables. We find that performance on all but the most trivial tasks suffered for both single digit and BitToken encoding strategies when including multi-step operations. As expected, the performance loss was lowest for the lowest cap of 0.1, but the resulting models are still not able to solve mean, standard deviation and exponentiation. While this performance improves with increasing cap size, it still does not reach the performance of the respective solo-task variants. Crucially, increasing the ratio of multi-step tasks results in a significant reduction of the ability to solve all other arithmetic tasks.
>
> Prospectively, we hypothesize this could be solved with sufficient scale. If an algorithm for the four elementary operations can be learned, as we have shown in this work, then, theoretically, a multistep algorithm that just requires multiple applications of the elementary algorithms can be learned with sufficient network depth. If this is computationally realistic remains to be seen. Alternatively, one could experiment with generating intermediate tokens to potentially unlock new algorithms while still maintaining a large efficiency advantage over single digit approaches.

---

> ### Author Response · Authors · 2025-11-20
> **Input precision versus model precision (W3)**
>
> We believe there may be a misunderstanding concerning precision. Our BitTokens represent input numbers as a 64-entry long vector. The individual vector elements correspond to 1 sign bit, 11 exponent bits, and 52 mantissa bits. This reflects the IEEE 754 definition of the float64 representation and allows encoding a maximum finite value of $\sim 1.7977 \times 10^{308}$ and minimum positive normal value of $\sim 2.225 \times 10^{-308}$ with 15-17 digits of precision.
>
> However, the individual entries of this vector can be encoded in a different precision, such as FP32, FP16, or FP8, without affecting BitToken’s ability to encode numbers. This is because BitToken encodings only require a single bit per entry to uniquely encode its entire representational range. This becomes important in practice, because the precision of the activations of an LLM determine the precision of the intermediate token representations. When a token enters a layer of a transformer, it is cast to the precision of the compute kernel which is typically determined by the precision of the activations.
>
> In our experiments, our transformer network’s weights and activations were trained using PyTorch’s automatic mixed precision. This means most activations are in float16. The reviewer rightly mentions FP8, which has recently emerged as a further level of precision reduction for faster LLM training and inference. However, we did not explicitly test this in our work as our networks use flash attention, which currently requires H100 GPUs for FP8 training, to which we had no access.
>
> We have updated the sixth desideratum to more clearly reflect that it refers to the use of low-precision network activations:
>
> > Numerical stability: Representations remain accurate when using low-precision network activations (e.g., FP8).
>
> We believe that compatibility with low-precision activations is highly desirable for number encodings considering the current trend of hardware and research to support quantization to FP8 [1][2], with DeepSeek V3 [3] being trained in mixed FP8. Methods that require higher precision to differentiate between numbers, such as xVal, would be unable to faithfully generate intermediate representations during the forward pass, and are hence incompatible with such advances.
>
> [1] Fishman, Maxim, et al. "Scaling fp8 training to trillion-token llms." ICLR (2025).
>
> [2] Ma, Shuming, et al. "The Era of 1-bit LLMs: All Large Language Models are in 1.58 Bits." arXiv preprint arXiv:2402.17764 (2024).
>
> [3] Liu, Aixin, et al. "Deepseek-v3 technical report." arXiv preprint arXiv:2412.19437 (2024).

---

> ### Author Response · Authors · 2025-11-20
> **Additional ablation experiments (W4) (1/2)**
>
> We have added most of the ablations suggested by the reviewer to Appendix D.3. These have shown that for the choice of the number head, a linear layer, no head (i.e. directly decoding from the last hidden layer), and a 2-layer multi-layer perceptron with ReLU non-linearity perform very similar:
>
> | | Metric | None | Linear | MLP |
> | :--- | :--- | :---: | :---: | :---: |
> | Min/Max | $\uparrow$ Exact match acc | 0.999 | **1.000** | 0.998 |
> | Interval | $\uparrow$ Exact match acc | **1.000** | **1.000** | **1.000** |
> | Sorting | $\uparrow$ Exact match acc | **0.994** | 0.992 | 0.989 |
> | Add | $\uparrow$ $\mathrm{log\textnormal{-}sMAPE}$ | 0.985 | **0.990** | 0.989 |
> | Mult | $\uparrow$ $\mathrm{log\textnormal{-}sMAPE}$ | 0.993 | 0.992 | **0.994** |
> | Div | $\uparrow$ $\mathrm{log\textnormal{-}sMAPE}$ | 0.993 | 0.981 | **0.995** |
> | Fineweb | $\downarrow$ Perplexity | 47.69 | 48.29 | **47.50** |
> | Fineweb text only | $\downarrow$ Perplexity | 51.08 | 51.91 | **51.06** |
>
> We also find that exchanging RMS norm for layer norm does not particularly affect performance, resulting in very slight decreases across almost all tasks:
>
> | | Metric | Layer norm |
> | :--- | :--- | :---: |
> | Min/Max | $\uparrow$ Exact match acc | $0.997^{\color{red}{-0.003}}$ |
> | Interval | $\uparrow$ Exact match acc | $1.000^{\color{gray}{+0.000}}$ |
> | Sorting | $\uparrow$ Exact match acc | $0.976^{\color{red}{-0.016}}$ |
> | Add | $\uparrow$ $\mathrm{log\textnormal{-}sMAPE}$ | $0.981^{\color{red}{-0.009}}$ |
> | Mult | $\uparrow$ $\mathrm{log\textnormal{-}sMAPE}$ | $0.988^{\color{red}{-0.004}}$ |
> | Div | $\uparrow$ $\mathrm{log\textnormal{-}sMAPE}$ | $0.987^{\color{green}{+0.006}}$ |
> | Fineweb | $\downarrow$ Perplexity | $50.10^{\color{red}{+1.81}}$ |
> | Fineweb text only | $\downarrow$ Perplexity | $53.94^{\color{red}{+2.03}}$ |
>
> The effects of different data mixing ratios when including the multi-step operations during multi-task training have already been discussed in our response to the reviewer’s third question. Additionally, we have now conducted experiments with different task sampling strategies. As explained in Appendix C.4, sampling is influenced by a scaling factor $\lambda$ that controls how the performances across the various tasks are averaged. We compared our default aggressive oversampling for low performing tasks using the harmonic mean ($\lambda=-1$), moderate oversampling using the geometric mean ($\lambda=0$), and performance independent sampling using the arithmetic mean ($\lambda=1$). We find that the harmonic mean used throughout the paper performs best:
>
> | | Metric | harmonic | geometric | arithmetic |
> | :--- | :--- | :---: | :---: | :---: |
> | Min/Max | $\uparrow$ Exact match acc | **1.000** | 1.000 | **1.000** |
> | Interval | $\uparrow$ Exact match acc | **1.000** | **1.000** | **1.000** |
> | Sorting | $\uparrow$ Exact match acc | 0.992 | 0.995 | **0.997** |
> | Add | $\uparrow$ $\mathrm{log\textnormal{-}sMAPE}$ | **0.991** | 0.983 | 0.710 |
> | Mult | $\uparrow$ $\mathrm{log\textnormal{-}sMAPE}$ | **0.995** | 0.990 | 0.979 |
> | Div | $\uparrow$ $\mathrm{log\textnormal{-}sMAPE}$ | 0.981 | **0.995** | 0.978 |
> | Fineweb | $\downarrow$ Perplexity | **48.29** | 53.13 | 77.28 |
> | Fineweb text only | $\downarrow$ Perplexity | **51.91** | 57.25 | 83.85 |

---

> ### Author Response · Authors · 2025-11-20
> **Additional ablation experiments (W4) (2/2)**
>
> According to the suggestion of the reviewer we have trained variants of all models using the AdamW optimizer. In order to achieve a competitive performance, we had to test several learning rates for each baseline. We find that the performance for all five number encoding strategy decreases compared to their counterparts trained with the Muon optimizer, with varying level of degradation. Crucially, our BitToken model trained with the Muon optimizer achieves the best performance across all settings.
>
> We want to emphasize that our decision to use Muon was based on the official recommendations from the maintainers of the modded-nanoGPT2 LLM that we are using for all our experiments [1]. Moreover, Muon has seen increased adaptation, including in training of frontier LLMs such as Kimi V2 [2], owing to its increased efficiency and performance [3].
>
> | | Metric | Subword | Single Digit | xVal | FoNE | BitToken |
> | :--- | :--- | :---: | :---: | :---: | :---: | :---: |
> | Min/Max | $\uparrow$ Exact match acc | $0.988^{\color{red}{-0.003}}$ | $\mathbf{0.989}^{\color{red}{-0.002}}$ | $0.001^{\color{gray}{+0.000}}$ | $0.842^{\color{red}{-0.017}}$ | $0.988^{\color{red}{-0.012}}$ |
> | Interval | $\uparrow$ Exact match acc | $\mathbf{0.994}^{\color{green}{+0.001}}$ | $0.989^{\color{red}{-0.003}}$ | $0.308^{\color{red}{-0.016}}$ | $0.994^{\color{red}{-0.003}}$ | $0.991^{\color{red}{-0.009}}$ |
> | Sorting | $\uparrow$ Exact match acc | $0.923^{\color{red}{-0.004}}$ | $\mathbf{0.928}^{\color{red}{-0.027}}$ | $0.000^{\color{gray}{+0.000}}$ | $0.530^{\color{red}{-0.044}}$ | $0.911^{\color{red}{-0.081}}$ |
> | Add | $\uparrow \text{log-sMAPE}$ | $0.942^{\color{red}{-0.012}}$ | $\mathbf{0.951}^{\color{red}{-0.023}}$ | $0.097^{\color{green}{+0.007}}$ | $0.805^{\color{red}{-0.004}}$ | $0.899^{\color{red}{-0.091}}$ |
> | Mult | $\uparrow \text{log-sMAPE}$ | $0.436^{\color{red}{-0.014}}$ | $\mathbf{0.640}^{\color{red}{-0.253}}$ | $0.098^{\color{green}{+0.004}}$ | $0.255^{\color{red}{-0.045}}$ | $0.582^{\color{red}{-0.410}}$ |
> | Div | $\uparrow \text{log-sMAPE}$ | $0.509^{\color{red}{-0.013}}$ | $0.482^{\color{red}{-0.199}}$ | $0.096^{\color{red}{-0.003}}$ | $0.238^{\color{red}{-0.037}}$ | $\mathbf{0.596}^{\color{red}{-0.385}}$ |
> | Fineweb | $\downarrow$ Perplexity | $78.08^{\color{red}{+3.25}}$ | $74.46^{\color{red}{+0.37}}$ | $134.07^{\color{red}{+31.50}}$ | $83.46^{\color{red}{+3.68}}$ | $\mathbf{74.32}^{\color{red}{+26.03}}$ |
> | Fineweb text only | $\downarrow$ Perplexity | $\mathbf{79.89}^{\color{red}{+5.06}}$ | $83.30^{\color{red}{+6.74}}$ | $146.83^{\color{red}{+31.50}}$ | $91.09^{\color{red}{+4.08}}$ | $80.29^{\color{red}{+34.80}}$ |
>
> The reviewer suggests examining the effect of lowering the precision of the input numbers during arithmetic benchmarking. We believe a major strength of our benchmark is that it tests float64 numbers, which provide a representational capacity and range not tested anywhere else in the literature. Being able to effectively operate upon float64 numbers enables language models to solve most tasks in mathematics, science, and engineering. As such, it must be an essential property of any number representation.
>
> The reviewer also mentions noise robustness as a possible ablation experiment. There is no reason to add noise to the input numbers since all evaluated methods deterministically convert a number into tokenized encodings. The outputs of the network are already subject to stochastic noise due to the fact that we are training a deep neural network. Thus, we believe that our experimental setup already comprehensively tests the ability of  number encodings to handle stochastic noise. However, we would gladly expand this analysis, provided the reviewer has a particular experimental setup in mind.
>
> [1] Jordan et al. "modded-nanogpt: Speedrunning the nanoGPT baseline." https://github.com/KellerJordan/modded-nanogpt, (2024).
>
> [2] Kimi Team, et al. "Kimi k2: Open agentic intelligence." arXiv preprint arXiv:2507.20534 (2025).
>
> [3] Jordan et al. "Muon: An optimizer for hidden layers in neural networks." https://kellerjordan.github.io/posts/muon/, (2024).

---

> ### Author Response · Authors · 2025-11-20
> **Experiments using production-scale LLMs (W5)**
>
> While conducting our study, we explored the possibility to extend our experiments to include larger datasets and language models. In particular, we considered training the fully open OLMo LLM [1] with different tokenization strategies. However, we calculated that training the smallest 1B parameter OLMo variant from scratch would require 26,752 H100 GPU hours, or roughly 4.5 months of training on a standard 8-GPU rack. This number has to be multiplied by the number of baselines and does not include any computational resources required for hyperparameter tuning or debugging. These resource requirements made it all but infeasible for us to explore the scaling properties of our proposed method to production-scale LLMs.
>
> We understand that only focusing on smaller models trained from scratch is a limitation of our work as it currently stands, but we strongly believe that our frontier model analysis, extensive theoretical analysis, and strong results of our novel method using the well researched, established nanoGPT2 architecture are all very valuable contributions to ICLR. Through the publication of this work, which contains an extensive description of our method as well as accompanying public code, we enable other researchers to potentially adapt and refine our method, and ultimately integrate single-token number encodings into larger LLMs.
>
> [1] Team OLMo, et al. "2 OLMo 2 Furious." arXiv preprint arXiv:2501.00656 (2024)

---

> > ### Comment · Reviewer_QH8n · 2025-11-20
> > **Continual pretraining?**
> >
> > Thanks for the response. I think we shouldn't aim for pretraining from scratch but at least we can do some continual pretraining with BitTokens? That being said, we take a reasonably good small model such as Llama 3.1 8B or Qwen3 models, replace their number embedding techniques with BitTokens, and continual pretrain on some text/number corpus or the multi task setup as your current version, and then evaluate on mathematical reasonings such as simply GSM8K to see if BitTokens could help in such scenarios? Let me know if this makes sense.

---

> > > ### Author Response · Authors · 2025-12-03
> > > **The benefit of controlled experiments from scratch**
> > >
> > > We appreciate the reviewer’s suggestion and agree that the suggested experiment would be an exciting direction for future work. However, there are several reasons we believe that this experiment is beyond the scope of the current paper.
> > >
> > > On the one hand, continual pretraining with a modified vocabulary introduces significant risk of catastrophic forgetting, especially when altering a core capability such as number representation. Experiments at this scale would require multiple specialized training datasets, careful monitoring of LLM training dynamics, and comprehensive evaluation across multiple diverse tasks.
> > >
> > > We would like to emphasize that we deliberately chose controlled experiments on small language models, as suggested in [1], to isolate the effect of number encoding strategies. This design ensures that observed differences between encoding methods are attributable to the method rather than confounding factors such as pretraining quality or fine-tuning instability.
> > >
> > > On the other hand, our current manuscript already delivers substantial contributions:
> > > - An evaluation of state-of-the-art LLMs, highlighting the reasoning cost of common arithmetic operations,
> > > - A set of desiderata for evaluating number encoding strategies,
> > > - A theoretic analysis revealing the limitations of existing single-token approaches, and
> > > - BitTokens, a novel encoding strategy that consistently outperforms current state-of-the-art methods by up to 30 points, allowing even small language models to solve basic arithmetic without the need for reasoning chains.
> > >
> > > We believe these contributions are both significant and valuable to the community. Adding a finetuning experiment as a side note would not do justice to its complexity and would be better suited as a standalone paper.
> > >
> > > [1] Ye, Tian, et al. "Physics of language models: Part 2.1, grade-school math and the hidden reasoning process." arXiv preprint arXiv:2407.20311 (2024).

---

> ### Author Response · Authors · 2025-11-20
> **Minor differences in perplexity scores (W6, Q1)**
>
> In response to the reviewer’s comments regarding the reported solo-task perplexity performance, we have conducted a series of additional experiments. Originally, we computed all fineweb perplexity scores on the same small validation subset. To investigate the differences in perplexity scores pointed out by the reviewer, we have recalculated the perplexity on five independent subsets of the FineWeb validation dataset. The results, shown in the table below, highlight that any differences between our proposed method and the four evaluated baselines are minor and do not exceed the measured standard deviation.
>
>
> | Small sample   | Subword       | Single Digit  | xVal          | FoNE          | BitToken      |
> |----------------|---------------|---------------|---------------|---------------|---------------|
> | PPL            | 32.90 +- 0.84 | 30.51 +- 0.88 | 29.72 +- 0.81 | 28.76 +- 0.79 | 28.27 +- 0.78 |
> |  PPL text only | 33.44 +- 0.77 | 33.33 +- 0.80 | 31.81 +- 0.71 | 30.85 +- 0.69 | 30.33 +- 0.69 |
>
> The observed dependency of perplexity on the subset used for evaluation prompted us to recalculate all reported scores using a substantially larger partition of 10,000 samples of the FineWeb text corpus. We consequently have updated all the perplexity scores in our paper. However, so far we have not observed any meaningful changes in the ranking of the evaluated methods and drawn conclusions.
>
> | Large sample   | Subword | Single Digit | xVal  | FoNE  | BitToken |
> |----------------|---------|--------------|-------|-------|----------|
> | PPL            | 32.56   | 30.39        | 29.65 | 28.67 | 28.33    |
> |  PPL text only | 33.13   | 33.29        | 31.62 | 30.65 | 30.19    |
>
> Additionally, we calculated a text-only perplexity score that excludes any numeric tokens from the evaluation, as suggested by the reviewer. This aims to reduce any effects resulting from the various number encoding strategies that use different amounts of tokens to represent numbers. As before, the performance differences between all methods is very small.
>
> Because of these observations, we would caution against too strong interpretations of small differences in perplexity on a single dataset. As demonstrated above perplexity is strongly influenced by the size and composition of the testing data set. In our interpretation, all evaluated models exhibit comparable ability to process language. Finally, we would also like to underline that even if FoNE has minimally lower complexity in the single task setting, its poor performance on arithmetic precludes it from being an effective number encoding.

---

> ### Comment · Reviewer_QH8n · 2025-11-20
> **clarification of my question**
>
> Thanks for the clarification. For this particular question, I was meant to ask, for any number $x$, if you use `float32(x)` embedding instead of `float64(x)` (what the current method is using), would that change the model performance significantly? Interestingly as well, if you use `bfloat16(x)` or even lower precisions as the number embedding, how much would that change the performance. In short, my question is, is float64 necessary? If we can't prove this necessity together with its sufficiency, why don't we use higher or lower precision? Hope this clarifies my question better!

---

> > ### Author Response · Authors · 2025-12-03
> > **Performance for float32**
> >
> > We would like to thank the reviewer for the clarification. As we understand the reviewer, their question consists of two parts:
> > 1. Is 64-bit precision (≈15-17 significant digits) truly necessary for real-world applications, or would lower precisions such as 32-bit (≈6-9 digits) or even bfloat16 (≈2-3 digits) suffice?
> > 2. If 6 digits of precision (float32) would be sufficient, how would our proposed encoding strategy as well as the evaluated baselines perform on this simpler task?
> >
> > Our choice of float64 was motivated by both theoretical and practical considerations. While many prior works focus on arithmetic tasks involving relatively small ranges (3–6 digits), which can be represented accurately with float32, real-world scenarios often require higher precision. However, float32 numbers can only guarantee the correctness of 6 significant digits. For example the Python statement
> > `np.float32(8_589_973.0) + np.float32(0.5)` results in the number `8_589_974.0`.
> > Such rounding errors accumulate when chaining operations, producing incorrect results. Lower precisions exacerbate this issue. For example torch.bfloat16 cannot even represent the integer 257.
> >
> > This precision matters when solving real-world problems in many domains. Consider these two examples:
> > - Compound interest: Imagine a savings account with an initial deposit of `$1,000,000`, an annual interest rate of 5%, compounded daily for 10 years. The final amount is computed via $A=P\times (1+\frac{r}{n})^{n \times t}$, where P=1,000,000, r=0.05, n=365, t=10. In float64, the final amount is `$1,648,664.81`, but in float32 it is `$1,648,575.50`. Over millions of accounts, these errors could scale to millions of dollars.
> > - Science: many constants in physics and chemistry such as the speed of light (c = 299,792,458 m/s) and Planck's constant: (h = 6.626070150e-34 J$\cdot$s) would lose precision and thus result in incorrect calculations. For example, defining the speed of light in float32 fails because the format lacks the precision to hold all nine digits, silently rounding it to 299,792,448. This 10 m/s error accumulates rapidly: in a simple one-hour satellite trajectory calculation using the doppler shift with d = c * t, your position would be wrong by 36 kilometers.
> >
> > This illustrates that 64-bit precision is necessary for real-world applications. However, to address the reviewer’s second point, we created a 32-bit version of our dataset with the same distribution as the original one and trained all models under this reduced precision.
> >
> >
> > |                           | Metric            | Subword | Single digit | xVal   | FoNE  | BitToken  |
> > |---------------------------|-------------------|---------|--------------|--------|-------|-----------|
> > | Min/Max                   | ↑ Exact match acc | 0.990   | 0.990        | 0.002  | 0.807 | **1.000** |
> > | Interval                  | ↑ Exact match acc | 0.996   | 0.998        | 0.374  | 0.997 | **1.000** |
> > | Sorting                   | ↑ Exact match acc | 0.955   | 0.957        | 0.0    | 0.517 | **1.000** |
> > | Add                       | ↑ log-sMAPE-32    | 0.984   | 0.987        | 0.230  | 0.916 | **0.994** |
> > | Mul                       | ↑ log-sMAPE-32    | 0.874   | 0.983        | 0.235  | 0.638 | **0.998** |
> > | Div                       | ↑ log-sMAPE-32    | 0.883   | 0.960        | 0.235  | 0.638 | **0.998** |
> > | Fineweb                   | ↓ Perplexity      | 55.36   | 50.09        | 112.52 | 67.02 | **43.68** |
> > | Fineweb text only         | ↓ Perplexity      | 56.40   | 55.56        | 123.13 | 72.83 | **46.86** |
> > | Fineweb numeric text only | ↓ Perplexity      | 50.06   | 49.62        | 117.69 | 66.36 | **41.22** |
> >
> >
> > The table above demonstrates that reducing the difficulty of the tasks leads to higher accuracy for all models. Notably, even for these lower precision requirements, there are still samples across all tasks that the baselines are not able to solve reliably, highlighting the benefit of our proposed BitToken strategy.

---

> ### Comment · Reviewer_QH8n · 2025-11-21
> **re: solo task's perplexity**
>
> I think for perplexity, we should evaluate perplexity of text tokens, only on sequences that contain both numbers and texts, to show that the proposed method could lead to better understanding of numeracy. Looking at all sequences will dilute the insights, even with the updated ppl text only (which of course is better than purely ppl on all tokens).

---

> > ### Author Response · Authors · 2025-12-03
> > **Text only perplexity scores for sentences with numbers**
> >
> > As the reviewer suggested, we evaluated all models once again on the same large validation dataset, but now only compute the loss for tokens that are within a range of 20 tokens before or after a numerical token. The obtained results clearly show that the ranking of the models remains the same as in our previous results.
> >
> > |                           | Subword | Single digit | xVal  | FoNE  | BitToken  |
> > |---------------------------|---------|--------------|-------|-------|-----------|
> > | Fineweb                   | 32.56   | 30.39        | 29.65 | 28.67 | **28.33** |
> > | Fineweb text only         | 33.13   | 33.29        | 31.62 | 30.65 | **30.19** |
> > | Fineweb numeric text only | 28.61   | 28.99        | 27.60 | 26.75 | **26.03** |

---

### Official Review · Reviewer_bJvs · 2025-11-01

**Soundness:** 2
**Presentation:** 3
**Contribution:** 2
**Rating:** 4
**Confidence:** 3

**Summary:**

The manuscript addresses the problem of LLM performance on arithmetical operations over numerical expressions. It identifies shortcomings of current implementations supported by empirical evidence, critically surveys alternative solutions available in the state of the art, lists a series of desiderata, and proposes a novel alternative accompanied by experimental results. In my view, the manuscript exhibits clear strengths but also important weaknesses, suggesting significant room for improvement.

**Strengths:**

1. The problem of arithmetical content processing in distributional models, and LLMs in particular, is an important and timely one
2. The manuscript is well-written and well-structured
3. It provides a reasonable account of the literature relevant to the solution proposed, even if some relevant work is missing (cf. weaknesses)
4. Good and informative empirical evaluation on frontier LLMs
5. Very interesting analysis of addition and multiplication over sinusoidal encoding of numerical expressions
6. Empirical and formal results are sound as far as I can judge (disclosure: I'm not a mathematician by training, so, despite my best efforts, it is not impossible that I have overlooked some technical details)

**Weaknesses:**

(in order of importance)
1. The motivation for this work is ill-formed, and therefore, the solution proposed can look unjustified
2. The experimental results obtained are relatively limited
3. Desiderata look arbitrary
4. Using prompting as a method is inadequate
5. Relevant work is not considered
6. Confusion between tokenization and embedding
7. Lack of clarity in formal statements

Further details on weaknesses:
1. The problem of the arithmetical performance of LLMs is presented in terms of "numeracy" of LLMs, defined as "the ability to understand and work with numbers." As such, this problem is conceived as a cognitive task. But the problem of arithmetical operations in LLMs, no less than in an elementary pocket calculator, or any abstract or concrete computational model for that matter, is not a cognitive but a formal one. My calculator---or, say, the lambda calculus---doesn't have "the ability to understand and work with numbers", while one can arguably say that my neighbor does. Certainly, we have some good ideas of why my calculator performs arithmetical operations, while my neighbor's cognitive capabilities are more obscure. But the cognitive character of the latter doesn't come from that obscurity. Therefore, the fact that the operations of a computational model like an LLM are obscure to us is not a legitimate reason to assume that whatever numerical calculation they perform is to be addressed in cognitive terms as "numeracy". These remarks are not purely speculative. They point to the fact that attempting to provide an LLM with "intrinsic numeracy skills" is an ill-defined task at best (if not belonging to magic altogether). With respect to LLMs, the problem of numerical calculation is either a descriptive one (i.e., understanding the formal mechanisms explaining the possibilities and limitations of distributional models of computation to perform arithmetical calculations) or a normative one (i.e., what alternative mechanisms can we imagine for correct calculations). This manuscript provides some interesting insights on the former by formally analyzing alternative solutions (in particular, sinusoidal encoding, which, however, is not distributional). But if the problem is how to enhance distributional models with formal methods for correct calculation, it is not clear why not simply outsource calculations to an elementary calculator, which would be, without any doubt, more efficient and more effective. The manuscript claims that this "prevents the model from building an intuition for numbers and the results of calculations, which is required to interpret and contextualize information from complex domains", which again, I take for highly unscientific, since "intuition", "interpretation", and "contextualization" are not computational concepts. One could maybe claim that outsourcing numerical calculations affects the processing of non-numerical (eg. linguistic) expressions in LLMs, but this would require evidence that neither is present in this manuscript, nor seems to be its intention to provide. One could also claim that outsourcing calculations would interfere with the current tendency of constructing end-to-end models, but the proposed solution is no better in this sense, because it comes down to introducing symbolically engineered components foreign to the end-to-end distributional approach, and yet are less efficient and perform worse than a simple numerical calculator. Another way to put it is: why should numerical calculations be trainable in an LLM, once acknowledged that, one one side, training methods are highly inefficient and ineffective, and on the other, we have simple, well-understood, very efficient, and 100% correct formal methods for numerical calculation available in case we are ready to enhance a trained model with something else? Without an answer to this question, there is a risk of not addressing the problem at hand (numerical calculation in LLMs) with the right tools, introducing spurious concerns, while neglecting important dimensions. I believe that, due to the questionable motivation, the manuscript suffers from both (see other points below).
2. Even if one disregards the motivations, one could claim that the results of the proposed method are relatively modest, with significant improvements with respect to the leading baseline only for multiplication and division, while performing significantly worse on the computation of the mean, and slightly worse on 3 other of the 8 tasks. This wouldn't be a problem *per se*, if these results were mobilized for descriptive purposes, giving solid insights on the mechanisms responsible for the different behaviors, instead of evidence for a normative goal (i.e., making the models better). However, the manuscript only reports the performance of varying representation strategies (Appendix D.3) without analysis of the possible reasons behind the difference in performance. And for the interesting case of the mean, it attributes the advantage to other methods, without evidence, to "the fact that multi-token methods generate answers over multiple forward passes, which effectively enables a form of “reasoning”", once again hiding behind cognitive metaphors the lack of understanding of the corresponding formal mechanisms.
3. The desiderata advanced in section 3, supposed to justify the proposed methods, are introduced without sufficient discussion and, therefore, appear as a more or less arbitrary list of properties matching the solution (and not the other way round) instead of a coherent system of independent conditions (as in an axiomatic system), especially if one can raise doubts about the overall motivation of the paper, as discussed in the point 1 above. As an example, one could claim, unlike D1, that representing numbers with as many tokens as digits for some positional representation in some base (eg. representing the number 123 with the 3 consecutive tokens "1" "2" "3") is the most efficient way of representing numbers, making algorithmic properties readily available to computation (hence the importance of positional numerical systems since the Babylonians). Or that making numbers independent of geometry, unlike D3, actually frees arithmetic from geometric limitations, as centuries of abstract algebra have shown. All this can be debated, and depends on what is the point of computing arithmetical properties in one way or another, but the point is that there is nothing obvious in the desiderata proposed, which would then require further justification.
4. A rigorous approach to understanding how LLMs perform arithmetical computation should not accept prompting (e.g., "you are an expert in numeracy... do not explain...") as a valid method, as none of this has a rigorous computational/formal status. I know this is widespread practice in the field, but I don't think that's a legitimate argument. I also know that not all models are open source, and there's no other way to explore them than prompting. But that's not a reason to accept prompting as a scientific methods, but a reason not to use those models for scientific purposes. I'm reviewing a scientific paper, not a commercial product. Can you be sure that a model didn't use a calculator tool just because you asked it to please not do it? If that can't be guaranteed formally, then any result coming from such an obscure procedure falls necessarily outside the domain of computer science and belongs to other areas of scientific knowledge, such as anthropology or psychology, or non-scientific, such as religion, or magic. Not having a clear motivation, as pointed in 1., hides the problem with this methodology.
5. Since a central aspect of this paper has to do with the representation of numerical expressions and their mathematical processing in the framework of transformer models, it is surprising not to see any discussion of the work done by Charton on this topic ([eg1](https://arxiv.org/pdf/2308.15594), [eg2](https://arxiv.org/pdf/2306.15400),[eg3](https://arxiv.org/pdf/2211.00170), [eg4](https://arxiv.org/pdf/2112.01898)). Discussing his views and adopting some of his solutions could be precious for the work presented in this manuscript (Disclosure: I am neither Charton, nor any one of his co-authors).
6. The manuscript presents the problem as a tokenization problem ("We hypothesize that addressing this problem requires rethinking the way LLMs tokenize numbers."; and BitTokens is presented as "a novel tokenization strategy"), but the problem is clearly not one of tokenization, but of embedding. From a tokenization perspective, the solution proposed is trivial: every numerical expression is mapped to the same [NUM] token (which, incidentally, exposes the LLM to the risk of statistical inconsistency, cf. [Gastaldi et al., 2025](https://arxiv.org/abs/2407.11606)). It was not until page 3 or 4 that I understood that tokenization was not the issue. I suggest to remove any substantial reference to tokenization and frame the paper in terms of encodings, vector representations, or embeddings.
7. Proposition 4.3 is expressed in rather informal terms ("numbers", "states" "outputs", "read", etc), in a way that it is difficult to follow the correctness of the proof proposed. For instance, it's not clear where the contradiction lies (because nowhere was explicitly claimed that the operator was injective), let alone the fact that a proof by contradiction might not be needed at all here, since an explicit bound is being found (e.g., a direct proof could claim that below that bound the operator is not injective, or sthg of the sort). The proof of the computational complexity is even less formal, so, while I think I understand the argument, I'm not sure I can follow the correctness of a proof. To avoid misunderstandings, I suggest either providing a more formal statement and proof, or presenting this more like an argument than like a formal result.

**Questions:**

- Would you be ready to change the motivation of the paper to avoid appealing to obscure cognitive properties? If you were not allowed to appeal to cognitive metaphors to justify your work, how would you justify the use of your method over outsourcing numerical computation to a simple numerical algorithm? Could you provide formal or empirical evidence for whatever that justification would be?
- Would you be ready to remove any claim about tokenization and frame this work exclusively in terms of encoding or embedding?
- What is your justification for considering it a rigorous scientific method to politely ask LLMs to do something? What alternative methods can you imagine to make your work more rigorous?
- What is the unified perspective that justifies the desiderata?
- It is unclear to me what the number sampling is supposed to reflect. Is it actual distributions on real-life corpora? Cognitive numeracy? Formal principles of learnability? How is the "increased difficulty" that justifies oversampling operands with similar exponents judged? Difficulty is usually a function of the algorithmic implementation, which is largely unknown in the case of DNNs. It could also be that by learning easy cases, a model generalizes better, and this is being artificially prevented by the sampling?

---

> ### Author Response · Authors · 2025-11-20
> **Arithmetic versus cognitive numeracy and the framing of our study (W1, Q1)**
>
> The reviewer makes the point that numeracy is a term that stems from the cognitive sciences, and is misplaced when discussing probabilistic computational models. We wholly agree with their perspective that LLMs are not cognitive beings and discussed at length internally if we should use the term numeracy at all. In the end, we opted to include it as we believe it provides an intuitive framework to discuss numeric ability. In particular, it introduces the concepts of numeric representation ability and numeric computational ability. However, the meaning of these concepts changes depending on whether they are discussed in the context of cognition or probabilistic language models.
>
> In the context of cognition, the concept of numeric representation ability is often framed as number understanding. This concept is only applicable to cognitive beings (such as the reviewer's neighbor) and not LLMs. We agree with the reviewer that such cognitive language is inappropriate for a computational model, thus we have removed all cognitive metaphors and anthropomorphisms from our paper. This includes but is not limited to terms such as *intuition*, *understanding*, and *interpretation*.
>
> However, in the context of probabilistic language models, we understand numeric representation ability to be a different concept. Here, it refers to how numbers are represented in inputs, outputs, and intermediate layers of the model. Encoding numbers in different ways leads to different representations that are more or less efficient for creating a probability distribution that will generate the correct next token. In this sense, it is still a valid concept and worth striving towards better numeric representations and thus better numeracy. We agree that our previous definition of numeracy was too imprecise in this regard and have adjusted it to:
>
> > Thus, to aid advancements in these fields, LLMs must possess efficient and effective numeracy, defined as the ability to represent and compute numbers.
>
> While we also hypothesize that our BitTokens are better numeric representations and have investigated this using our desiderata and targeted experiments, our main results concern numeric computational ability. We have changed many mentions of numeracy to "numeric computational ability" to reflect this focus.
>
> The reviewer also asks how we justify the need for our method, if our numeric computations can be done by a calculator more simply and with stronger guarantees as to correctness. While this is a fair question we believe our work is well motivated due to the following reasons:
>
> 1. Calculator calls always carry a computational and time cost. We aim to solve numeric computations correctly but also as efficiently as possible. Using our method, the computations are maximally efficient, as we solve the computation in a single forward pass. Any call to a calculator or other tool will require a formulation of the API call by the LLM (costing several tokens), followed by routing to an external service, waiting on the response, and insertion of the response. This cost will always exceed that of our approach, both in terms of computational resources and time.
> 2. Playing off of our previous point of numeric representations, calculators are not directly compatible with latent or intermediate calculations. The result of the computation must enter the network as a series of tokens and cannot be calculated dynamically during the forward pass to be used in intermediate layers.
> 3. There is academic interest in understanding if transformers can solve mathematical operations, as evidenced by a large body of work including that of Charton referenced by the reviewer.
>
> We have adjusted our introduction paragraph on calculators to better highlight their disadvantages:
>
> > Tool-augmented LLMs leverage external calculators or code to bypass the need for internal arithmetic computation (Schick et al., 2023; Qu et al., 2025; Gou et al., 2024; Zhang et al., 2023; He-Yueya et al., 2023; Parisi et al., 2022; Le et al., 2022; Wang et al., 2024b; Gu et al., 2024; Gao et al., 2023). While this approach guarantees the correctness of the calculations, it forces the model to dynamically and correctly identify, construct, and wait on all mathematical operations, introducing non-negligible latency and sources of error. Furthermore, outsourcing all calculations prevents the model from generating and calculating with intermediate numeric representations during a forward pass, restricting the efficiency of latent calculations (Skean et al., 2024; Hao et al., 2025; Lindsey et al., 2025).

---

> ### Author Response · Authors · 2025-11-20
> **Impact of our experimental results (W2)**
>
> We politely disagree with the reviewer's assessment that our results are only modest. On the four elementary operations, we achieve almost perfect results on tens of thousands of never before seen numbers and arithmetic problems across a wide range of difficulties. Single digit tokenization, the second-best performing method, underperforms our method by 10 and 30 points on multiplication and division, achieving only 0.893 and 0.681 log-sMAPE versus our 0.991 and 0.989. This is a clear indication of our BitTokens encoding approach making it feasible for even small transformers to learn some sort of general computational algorithm. While a full investigation of our models internal workings, for example using methods from mechanistic interpretability, is beyond the scope of this paper, we see no other feasible explanation for the observed generalization performance at this time.
>
> Regarding our diminished performance on the mean task, we believe the ability to solve the problem in multiple forward passes (such as single- & triple-digit tokenization strategies do) provides a large advantage in the algorithms that can be learned. This is what we mean by "reasoning". We recognize this is an imprecise cognitive term again and have removed it.

---

> ### Author Response · Authors · 2025-11-20
> **Justification for our desiderata (W3, Q4)**
>
> We appreciate that the motivation for the desiderata is very short. In response, we have expanded the introduction of Section 3. We have reincluded a description of the principles guiding the design of the desiderata, which was previously removed due to space constraints:
>
> > Our goal is to develop a number encoding strategy that generates representations which both maximize token efficiency and allow for neural networks to learn algorithms that perfectly execute numeric calculations. Such an algorithm or function must be injective and correct over the entire defined numeric input space. It also requires us to consider engineering realities, such as the normalization layers used, the numerical precision of the activation functions, and how gradient-based stochastic optimization learns representations.

---

> ### Author Response · Authors · 2025-11-20
> **Controlling the functioning of commercial LLMs (W4, Q3)**
>
> We share the reviewer’s view that fully open language models should be prioritized if we wish to advance scientific knowledge. On the other hand, current commercial models empirically perform the best across almost all tasks, and thus it would be remiss not to include them in our analysis. We acknowledge that we cannot know with 100% certainty how they have solved the posed problems, and that their performance should be taken as a possible upper-bound considering they could still be using calculator calls in the background.
>
> However, we would like to make the point that there is strong evidence that the commercial models do indeed follow our instructions. If they were making calculator API calls in the background despite explicit instructions not to, their capabilities while using calculators would be rather disappointing. None of the models achieve 100% performance on any of the arithmetic tasks despite it becoming trivial with calculator access. Furthermore, the models use (and bill us for) thousands of reasoning tokens, which clearly would not be needed if calculator calls were made. We also investigated the reasoning chains returned and saw the models note the fact that they are not allowed to use calculators. Inspections of individual chains also show that the final result of the model is achieved logically through the provided reasoning steps.
>
> We have included a strong caveat in the discussion of the results of the frontier LLMs that summarizes these arguments:
>
> > It is important to note that we have no way of controlling what the proprietary models (such as GPT 5 and Gemini 2.5) do during the generation of their answer. While we very explicitly instruct them not to use calculators, see that their reasoning chains align with non-calculator use, and they often do not answer correctly, we have no guarantee that they do not use external tools. Their reported results should be seen as an upper bound.

---

> ### Author Response · Authors · 2025-11-20
> **Further related work on transformer arithmetic including papers by Charton et al. (W5)**
>
> We appreciate the reviewer's recommendation of including Charton's work on transformer arithmetic. We originally placed our work solely in the literature of LLMs, but agree that investigations of the capabilities and limits of transformers to do math are also relevant. We have added a section in the introduction addressing this:
>
> > However, LLMs have historically struggled to solve even basic calculations. A growing body of work has thus tasked itself with finding the capabilities and limitations of transformers to perform mathematical operations [1,2,3,4].
>
> We also believe this body of work shows the importance of what we are working on, as they also recognize the potential of LLMs for science and math and the capabilities we need to fully unlock this potential. Similar to Charton's findings in [4], we also found that enforcing a log-uniform distribution of outcomes in the trainings set improves performance. We added this connection in the Appendix.
>
> > Similar to Charton (2024), we find that a log-uniform distribution of outcomes in the training set yields the best results.
>
> [1] Jelassi, Samy, et al. "Length generalization in arithmetic transformers." arXiv preprint arXiv:2306.15400 (2023).
>
> [2] Nogueira, Rodrigo Frassetto, Zhiying Jiang, and Jimmy Lin. "Investigating the Limitations of the Transformers with Simple Arithmetic Tasks." 1st Mathematical Reasoning in General Artificial Intelligence Workshop, ICLR (2021).
>
> [3] Lee, Nayoung, et al. "Teaching Arithmetic to Small Transformers." ICLR (2024).
>
> [4] Charton, François. "Learning the greatest common divisor: explaining transformer predictions." ICLR. 2024.

---

> ### Author Response · Authors · 2025-11-20
> **More precise use of the terms tokenization, encoding, and embedding (W6, Q2)**
>
> We fully agree that our core methodological contribution is the number encoding strategy and not the number tokenization strategy, which had already been suggested by previous works including xVal and FoNE. We have corrected this use of imprecise language throughout the manuscript, using tokenization only when appropriate and framing everything else in terms of encodings.
>
> We would like to emphasize that because each input number is mapped to a unique encoding corresponding to its unique float64 bit representation, as required by our uniqueness desideratum D2, there is no risk of statistical inconsistency as defined by Gastaldi et al. [1]
>
> [1] Gastaldi, Juan Luis, et al. "The foundations of tokenization: Statistical and computational concerns." ICLR (2025).

---

> ### Author Response · Authors · 2025-11-20
> **Stricter formalization of Proposition 4.3 (W7)**
>
> The reviewer correctly pointed out that our previous version conflated formal mathematical constructs with informal application language. We have substantially rewritten the proof to replace vague descriptions with precise mathematical definitions.
>
> In particular, we have added more details to the proof on the non-locality. While the operator $\otimes_\phi$ does not have to be injective, it must correctly calculate the output encoding components for all possible inputs. We show that if the operator accesses insufficient components, there exist distinct inputs $x_1,x_2$ that are indistinguishable by the operator. This leads to a contradiction, where the function is forced to produce identical outputs for products that should be distinct.
>
> As requested, we removed the vague discussion of disentanglement from the "computational complexity" section and further formalized the proof. We now formalize the encoding relationship as a linear system $\Theta=Mk \bmod 1$, where $M$ is a mixing matrix. We then utilize the Kronecker product $M\otimes_K M$ to model the information available in the entangled product space. This allows us to derive relative lower bounds on complexity. We prove that late disentanglement is algorithmically reducible to decoding since $(M\otimes_K M)^{−1}=M^{−1}\otimes_K M^{−1}$ and is numerically less stable, as quantified by the quadratic increase in the spectral condition number $\kappa(M)^2$.
>
> We believe that this improved formulation now provides greater formal rigor and presents our findings more clearly.

---

> ### Author Response · Authors · 2025-11-20
> **Rationale for distribution of our number dataset (Q5)**
>
> The main motivation of our number sampling is twofold. First, most numbers processed by scientists and engineers can be represented in float64. It is the de-facto standard of modern computing solutions and programming languages and if we can perform arithmetic perfectly in this space the LLM should be fully suited for scientific and engineering problems.
>
> Second, since the answer to our computation is also in float64, calculations become trivial if their exponents are farther apart than the precision limit of float64 (15-17 digits). For example adding a number with exponent $10^{5}$ to a number with exponent $10^{208}$ will not change the latter due to it being outside the precision limit of float64. Oversampling operands with similar exponents is essential to ensure that some calculation must be done to generate an answer, and it does not suffice to merely copy the large number. We also argue that most applied problems are based on numbers of similar magnitudes.
>
> Sufficient difficulty and a large enough problem space allows us to all but guarantee that the model must learn some general algorithm to achieve perfect accuracy on the breadth and depth of never-before-seen test samples. Approximations and interpolations would not suffice to achieve perfect generalization performance. This is the goal of our project and our sampling, and what we have achieved with our BitTokens.
>
> We have added this to Appendix A.2:
>
> > The difficulty of addition lies in the carry propagation. If the exponents of the operands are farther apart than the precision limit of float64, the answer to the calculation is simply the larger number. We therefore increase the number of samples where both operands are of similar magnitude.

---

### Author Response · Authors · 2025-11-20
**General response to all reviewers**

We would like to thank the four reviewers for their thorough review and helpful comments. We are encouraged that they find our work impactful, important, and timely. Moreover, the reviewers have positively commented on all three main contributions of our study: the empirical evaluation of frontier LLMs, formal analysis of single-token number encodings through the lens of nine desiderata, and our newly introduced BitTokens that demonstrably improve upon existing number encoding strategies.

In our rebuttal, we aim to address the reviewers' remaining concerns regarding the framing of our work, the guiding principles behind our desiderata, the experimental setup, and the interpretation of our results. To this end, we have individually responded to each of the reviewers' comments below, as well as implemented numerous changes in our manuscript, which are marked by blue font in the updated version.

Specifically, we have carefully revised how we motivate the need for single-token number encodings, culminating in the definition of our nine desiderata. Furthermore, we have included additional details on our methods and experiments, answering questions related to the generalizability and robustness of our method. Also, we have provided a clearer rationale for the introduction of an additional arithmetic benchmarking dataset and addressed concerns regarding the fairness of our experimental setup. We have also included several new ablation experiments. Finally, we have added an extended discussion on the necessary steps required to integrate our single-token number encodings with production-scale LLMs.

We believe that our work has substantially improved as a result of these changes, and invite the reviewers to comment whether we have adequately addressed their criticism. In case, there are any open questions or concerns, we are happy to answer further questions or discuss the scope of our work.

---

> ### Author Response · Authors · 2025-12-03
> **General response to all reviewers (2/2)**
>
> Although the rebuttal period has unfortunately been cut short, we believe that we were able to comprehensively address all the reviewers’ comments. In addition to our extensive rebuttal in the first round, we have addressed the remaining minor comments by reviewer QH8n and THn1.
>
> Specifically, we resolved the reviewers’ concerns regarding a potential training dataset disparity by adding a new ablation experiment. We also further justified our decision to evaluate on float64 precision by highlighting its necessity in numerous real-world applications, while also confirming that BitTokens maintain superior performance in a lower precision setting. Finally, we clarified some follow-up questions regarding the structure desideratum and the role of FP8 precision in modern LLMs.
>
> We would like to sincerely thank all the reviewers for their helpful feedback, and we find that our work has substantially improved as a result of the discussion. Overall, the discussion has helped us to sharpen our language and motivation throughout the paper, highlighting the potential impact of our research and underlining the necessity of our solution. Furthermore, the additional experiments and improved mathematical rigor helped us underline both the theoretical and empirical benefits of our approach.
>
> In closing, we would like to emphasize one last time the strengths of our paper that all reviewers have highlighted. Namely, that our work offers valuable insights into the limitations of arithmetic in even the most capable frontier LLMs, provides a practical and principled framework to analyze numeric encoding strategies through the lens of our desiderata, and establishes our BitTokens as a strong building block to advance capabilities of LLMs for scientific domains through their near perfect generalization performance on all four elementary arithmetic operations.

---

### Meta-Review · Area_Chair_T7GC · 2026-01-06

**Summary:**

This paper attempts to address a fundamental bottleneck in current LLMs: the efficient and accurate processing of numbers. By proposing "BitTokens"—a single-token embedding strategy based on the IEEE 754 floating-point standard—the authors offer a novel and principled alternative to existing methods like digit-by-digit tokenization, xVal, or FoNE.

Strengths:

1. Novelty & Principle: The core idea of aligning LLM number embeddings with the hardware-native binary representation (IEEE 754) is sound and innovative. The introduction of 9 formal desiderata provides a solid theoretical framework for evaluating number encodings.

2. Performance: BitTokens demonstrate superior performance on elementary arithmetic (addition, multiplication, division) compared to state-of-the-art baselines like xVal and FoNE, particularly in generalization to unseen numbers within the float64 range.

3. Efficiency: The method addresses the "token tax" of numbers in data-intensive applications (e.g., scientific or tabular data) by compressing numbers into single tokens without losing precision.

Rebuttal Quality: The authors provided a high-quality rebuttal that addressed critical concerns:

1. Fairness (Reviewer QH8n): They resolved concerns about unfair training distributions by demonstrating that models perform best on their native base distributions (binary vs. decimal) and that BitTokens maintain their lead even when baselines are favored.

2. Precision (Reviewer QH8n, THn1): They added experiments with float32 and clarified the compatibility with low-precision (FP8) activations used in modern frontier models (e.g., DeepSeek V3), strengthening the practical argument.

3. Framing (Reviewer bJvs): They successfully pivoted the terminology from cognitive "numeracy" to "numeric computational ability" and formalized their proofs, addressing presentation concerns.

Weaknesses:

1. Failure to Demonstrate Utility in Relevant Settings: The central promise of the paper is improving "numeracy in language models." However, the experiments are restricted to tiny models trained from scratch. The authors argue that training scale-relevant models (e.g., 7B parameters) is too expensive. While resource constraints are understandable, they do not justify creating a new encoding standard without verifying if it works in the actual setting it claims to improve (large-scale pre-training). There is no empirical evidence that BitTokens integrate well with the massive textual corpora used in modern LLMs without degrading general language capabilities or suffering from catastrophic forgetting, a critical concern raised by Reviewers QH8n and THn1.

2. Inherent Functional Limitations: The method trades flexibility for token efficiency in a way that hurts complex reasoning. As noted by Reviewers QH8n and bJvs, the model struggles significantly with multi-step operations like Mean, Standard Deviation, and Exponentiation compared to baselines. This confirms a major theoretical weakness: by compressing a number into a single token, the model loses the ability to perform "chain-of-thought" computation within the number's representation (e.g., carrying digits). In an era where scaling test-time compute is the primary driver of reasoning performance, a method that architecturally bottlenecks intermediate computation is a step backward.

Based on the above, a rejection is recommended at the current time. But I strongly expect to see improvements of the paper in future venues, particularly addressing: 1. how to combine single-token embedding's efficiency with long step-by-step reasoning's strengths for complex tasks, and 2. show promise on large-scale pretrained models and general reasoning tasks (even you cannot afford pretraining, you can give some recipes for companies or large labs to try, or you can do minimally affordable pretraining experiments to show your method is promising).

**Reviewer Concerns:**

Concerns addressed:

1. Experimental fairness (Reviewers QH8n, M71x, THn1): Reviewers suspected that BitTokens only outperformed baselines (like xVal and FoNE) because of unfair advantages in the training curriculum (binary vs. decimal difficulty sampling) or optimizer choice (Muon vs. AdamW). The authors conducted rigorous ablations, including training all models with AdamW (BitTokens still won) and testing cross-distribution training (training on binary difficulty then fine-tuning on decimal). They showed that models intrinsically prefer their native base distribution and that BitTokens' superiority holds even when these variables are controlled. They also reproduced FoNE's original results to prove their implementation was correct.

Outstanding concerns:

1. Generalization to reasoning & large models (Reviewers QH8n, THn1, M71x): Reviewers repeatedly asked how BitTokens affect performance on actual math word problems (e.g., GSM8K) or within large-scale pre-trained models (e.g., Llama/OLMo scale). The authors acknowledged this is a critical question but deemed it "out of scope" due to the prohibitive compute cost (estimated ~26k H100 hours) of training a large model from scratch. While the excuse is valid, the scientific question—does this actually help a general LLM do math better?—remains empirically unanswered.

2. Handling multi-step operations (Reviewer QH8n, bJvs): The method performs poorly on operations that require intermediate calculations (Mean, Standard Deviation, Exponentiation) because it compresses everything into a single token/forward pass, denying the model "reasoning tokens." While the authors explained why this happens, it remains a functional limitation compared to standard multi-token approaches that allow for chain-of-thought computation.

**Reviewer Scores:**

All the reviewers might slightly increase their scores. But I still hope to see such novel numerical embedding papers to show real impacts some day, otherwise they are only GPT-2 level toys that essentially shows using certain priors can help transformers learn certain algorithms/circuits. But whether they hurt general performance, whether they help pretrained general-purpose LLMs, are rarely answered.

---

### Decision · Program_Chairs · 2026-01-26

Reject